# Circulating microbial content in myeloid malignancy patients is associated with disease subtypes and patient outcomes

Jakob Woerner[1], Yidi Huang [1], Stephan Hutter [2], Carmelo Gurnari [3], Jesús María Hernández Sánchez[4], Janet Wang[1], Yimin Huang[1], Daniel Schnabel[1], Michael Aaby[1], Wanying Xu[1], Vedant Thorat [1], Dongxu Jiang[3], Babal K. Jha [3], Mehmet Koyuturk[5], Jaroslaw P. Maciejewski [3], Torsten Haferlach[2] & Thomas LaFramboise [1]✉

Although recent work has described the microbiome in solid tumors, microbial content in hematological malignances is not well-characterized. Here we analyze existing deep DNA sequence data from the blood and bone marrow of 1870 patients with myeloid malignancies, along with healthy controls, for bacterial, fungal, and viral content. After strict quality filtering, we find evidence for dysbiosis in disease cases, and distinct microbial signatures among disease subtypes. We also find that microbial content is associated with host gene mutations and with myeloblast cell percentages. In patients with low-risk myelodysplastic syndrome, we provide evidence that Epstein-Barr virus status refines risk stratification into more precise categories than the current standard. Motivated by these observations, we construct machine-learning classifiers that can discriminate among disease subtypes based solely on bacterial content. Our study highlights the association between the circulating microbiome and patient outcome, and its relationship with disease subtype.

[1] Department of Genetics and Genome Sciences, Case Western Reserve University, Cleveland, USA. [2] Munich Leukemia Laboratory, Munich, Germany. [3] Department of Translational Hematology & Oncology Research, Cleveland Clinic Foundation, Cleveland, USA. [4] Centro de Investigación del Cáncer, Salamanca, Spain. [5] Department of Computer Science, Case Western Reserve University, Cleveland, USA. ✉email: TXL80@case.edu

Myeloid malignancies are diseases that result from abnormal proliferation or lack of differentiation in myeloid progenitor cells. This class of neoplasms includes acute myeloid leukemia (AML) as well as other diseases that can progress to AML such as myelodysplastic syndrome (MDS), characterized by dysplastic changes of hematopoietic progenitor cells, and myeloproliferative neoplasm (MPN), an over-proliferation of cells. Patients with characteristics of both MDS and MPN are classified as MDS/MPN[1]. The annual incidence rate of myeloid malignancy is approximately 8 per 100,000 in Europe[2], for example, but is much higher among the elderly[3]. While survival has improved, it is highly variable among the different disease subtypes. In the US, AML 5-year survival is estimated at around 27%, and treatment options become increasingly limited and ineffective with increasing patient age[4]. A better understanding of the factors that influence disease outcomes and response to treatment is needed.

Meanwhile, evidence is growing for relationships between human cancers and the microbiome. In the blood cancer realm, some B-cell lymphomas have been associated with the bacteria *Helicobacter pylori*[5], while T-cell leukemia and Burkitt's lymphoma have long been known to be caused by viral infections (human T-cell lymphotropic virus 1[6] and Epstein-Barr virus[7], respectively). Recent work has investigated the relationships between the microbiome and clinical features in myeloid malignancy patients, though these studies have almost exclusively analyzed the gut microbiome. For instance, multiple studies have demonstrated that intestinal microbiota composition can predict survival in stem cell transplant patients[8,9]. The connection between the gut and the bone marrow is well-established[10], and therefore an impact of the gut microbiome on blood cancer and its treatment[11] is conceptually and empirically rational. However, the microbiome at the actual tumor site of myeloid malignancy—bone marrow and peripheral blood—remains unexplored. This stands in contrast to solid tumors, where microbiome research has recently been directed toward the tumor site itself[12–15]. A survey of >1500 tumors revealed distinct microbiome compositions for each of seven tumor types[15]. A similar study[16] of >18,000 solid tumor and matched normal blood samples was able to find microbial signatures in both the solid tissue and blood that could accurately predict tumor type, and the blood signatures could differentiate between cancer patients and healthy individuals.

Traditionally, human blood and bone marrow have been considered to be normally sterile, and therefore microbiome analysis of these entities would only be performed when deleterious infection was suspected. However, evidence is now accumulating for a normal blood microbiome in healthy individuals[17]. It is believed that microbiota in circulation is partially derived directly from the gut through bacterial translocation[18], and therefore the established ability of intestinal flora to predict patient outcomes may also be valid for blood. Given this and the increasingly acknowledged relationships between microbial communities and response to treatment, we hypothesized analogous relationships between the bone marrow microbiome in myeloid malignancy patients and disease characteristics. To this end, we extracted bacterial, fungal, and viral sequence from deep shotgun sequencing of DNA in the bone marrow and blood of 1870 myeloid malignancy patients, as well as in the bone marrow of healthy donors. Our primary goal was to elucidate relationships between microbial content/abundance and clinical features, including disease subtype and outcomes, in the disease cohort. Although our ability to perform case/control comparisons was limited by a relatively small number of healthy controls, we were nonetheless able to observe some significant differences in microbial content between patients and healthy donors.

## Results

**Patient overview and microbiome ascertainment.** The cohort comprised 1870 myeloid malignancy patients. Bone marrow ($n = 1756$) or peripheral blood ($n = 114$) was taken at diagnosis and sent to the Munich Leukemia Laboratory between 2005 and 2017 for work-up. Disease subtypes included AML ($n = 612$), MDS ($n = 640$), MDS/MPN ($n = 264$), and MPN ($n = 354$). Patient characteristics are provided in Table 1. Bone marrow samples from 12 healthy donors were also processed at the same site. DNA extracted from all samples was subjected to whole-genome sequencing, initially with the goal of comprehensively profiling somatic mutations in human DNA. Human genome average depth of coverage ranged between 76.8X and 183.8X (median 97.5X; Supplementary Data 1). We used PathSeq[19]—a tool that has been used in prior studies of tumor and blood microbes in cancer[16]—to identify reads derived from bacterial, fungal, and viral DNA. As shotgun metagenomic sequencing is known to be highly prone to artifacts (particularly for low-biomass samples such as blood), we followed strict filtering procedures. Briefly, two broad categories of reads were removed from consideration. First, we curated from the literature[16,20–23] a large list of 165 known problematic genera and 89 known problematic species (see the "Methods" section; Supplementary Data 2). Reads that aligned to any of these taxa were omitted from downstream analyses. Second, we manually inspected the breadth of genome coverage in species. Species showing read alignments only to very focal regions of their genomes indicate that the reads were derived from cryptic human sequences[20], and thus reads aligning to such species were also filtered out. These extremely strict filtering steps removed 184,919,804 of 185,938,531 reads (99.45%) mapping to unique genera and 128,135,955 of 129,323,801 reads (99.08%) mapping to unique

### Table 1 Patient characteristics.

| | n | Sex (% female) | Mean age (years) (1st, 3rd quartiles) | Karyotypic lesions (proportion) | | | | | | Median survival (years) |
|---|---|---|---|---|---|---|---|---|---|---|
| | | | | Normal | −5/−5q | −7 | Trisomy 8 | −Y[a] | Complex | |
| AML | 612 | 45.8 | 63.42 (54.38,75.20) | 0.367 | 0.077 | 0.088 | 0.103 | 0.066 | 0.203 | 1.41 |
| MDS | 640 | 42.7 | 71.07 (66.50,78.00) | 0.609 | 0.189 | 0.025 | 0.044 | 0.114 | 0.028 | 6.14 |
| MDS/MPN | 264 | 40.2 | 74.71 (70.78,80.72) | 0.712 | 0.008 | 0.042 | 0.131 | 0.045 | 0.027 | 5.36 |
| MPN | 354 | 36.7 | 62.79 (53.90,73.75) | 0.555 | 0.012 | 0.009 | 0.046 | 0.039 | 0.028 | 18.64 |
| overall | 1870 | 42.2 | 67.51 (60.85,77.10) | 0.533 | 0.095 | 0.046 | 0.076 | 0.074 | 0.086 | 5.71 |

[a]Calculated only from male patients.

species. Using the remaining reads, the burden of a taxon in a sample was quantified as the number of reads from the sample unambiguously aligning to the taxon, normalized to the number of human reads sequenced in the sample (see the "Methods" section).

**Microbial landscape differs between cases and controls and among disease subtypes.** The sequencing and filtering protocols yielded means of 48.6 fungal reads (s.d. 1408; range 0–56,363), 120.1 viral reads (s.d. 1588; range 0–27,420), and 3853.0 bacterial reads (s.d. 12,897; range 245–400,560) per sample (Fig. 1a). To visualize differences and similarities among the samples, we generated t-distributed stochastic neighbor embedding (t-SNE) plots from the relative abundances of genera (Fig. 1b), which showed clear grouping of normal control samples. All but one of the control samples have a first t-SNE coordinate >25, whereas only four of 1870 (0.2%) cases do (Fisher's exact $P < 2.2 \times 10^{-16}$). This suggests that microbe composition in the bone marrow of healthy individuals distinguishes them from disease cases. Furthermore, some grouping of patients by disease subtype was observed. For instance, more than half (52%) of MDS patients have second t-SNE coordinates below −10, while only 12.4% of patients from other subtypes did (Fisher's exact $P < 2.2 \times 10^{-16}$). These observations raise the potential for microbe content to differentiate among subtypes. Interestingly, in these t-SNE plots samples did not seem to group by age, sex, or blood/bone marrow status (Supplementary Fig. 7), suggesting that these factors are not strongly associated with microbial content. We did, however, observe some degree of temporal association with microbial content in the t-SNE plot (Supplementary Note). To ensure that this association did not cause artifactual results in the remainder of our study, we include supplementary analyses (see Supplementary Note) demonstrating that all results presented below remain valid even after controlling for temporal clusters.

We next calculated, for each pair of samples, the genus-level Bray–Curtis dissimilarity, which measures how different each pair is regarding microbial content. Normal controls are far more similar to one another than to the case samples (Wilcoxon $P < 2.2 \times 10^{-16}$; Fig. 1c). Interestingly, the same holds true for each disease subtype—that is, patients with the same subtype on average have lower dissimilarity than patients from different subtypes (Fig. 1d). We next computed the first two principal coordinates based on these dissimilarity measures. The resulting plots showed four distinct clusters (Fig. 1e). The four subtypes were not randomly dispersed among the four clusters. Instead, we observed strong enrichment in specific clusters for certain subtypes (Fig. 1f) (chi-squared test $P < 2.2 \times 10^{-16}$), providing further evidence that the microbial landscape may carry disease subtype-specific information in myeloid malignancy. Cluster membership also showed strong association with various karyotypic features, particularly normal karyotype, complex karyotype, and trisomy 8 (logistic regression $P = 0.00096$, $1.83 \times 10^{-6}$, and 0.023, respectively). For example, cluster 2 was enriched for normal karyotype patients, clusters 3 and 4 for complex karyotype, and cluster 1 for trisomy 8 (Fig. 1g). No relationship was observed between the clusters and either age or sex ($P = 0.10$ and 0.33, respectively; Supplementary Fig. 8).

**Human herpesviruses prevalent among myeloid malignancy patients and is associated with patient outcomes in MDS.** Although viruses have not been widely implicated in myeloid malignancies, they have been linked to patient outcomes and do have established roles in some lymphoid-lineage malignancies, as noted above. In our cohort, viral reads were detected in 1346 (72.0%) cases. Particularly prevalent were torque teno viruses,

which are extremely common in humans[24] and have very recently been implicated in a childhood APL case harboring a torque teno mini virus/RARA chimeric transcript[25], and human herpesviruses (Fig. 2a). In the latter category, *human betaherpesviruses 5* (human cytomegalovirus, HCMV) and *6* (roseolovirus), as well as *human gammaherpesvirus 4* (Epstein-Barr virus, EBV) occur at the highest burden. We observed extremely high levels of human betaherpesvirus 6 in nine (0.48%) patients. In these patients, the estimated number of copies of the viral genome approaches one per human cell. This observation is consistent with the known ability of human betaherpesvirus 6 to integrate into the human germline, which a recent study[26] found to occur in 0.58% of a cohort with primarily European ancestry. None of the other viruses showed evidence of germline integration. In contrast, in the normal controls viral sequence was detected only at very low levels and species could only be assigned in four of the 12 samples (Fig. 2b) (Wilcoxon $P = 0.019$ for difference in numbers of species-mapped viral reads in cases vs. controls).

Among cases, EBV and HCMV were by far the most frequently detected, found in 640 (34.2%) and 311 (16.6%) patients, respectively (Fig. 2c). The higher prevalence of these viruses in some of the patients may be a result of disease-related immunosuppression. Indeed, both EBV and HCMV are known to exist in a latent stage in large proportions of the population and can be reactivated upon suppression of host immune system[27]. Recognizing that our ability to detect the presence of a viral species may be strongly influenced by its genome length, we also recomputed prevalence for each species, adjusting for its genome length (see the "Methods" section). This adjustment recapitulates the dominance of EBV but shows a much-attenuated prevalence for HCMV (Supplementary Fig. 9).

Differences in overall viral read presence were observed among the four disease subtypes, with MDS patients showing the highest prevalence (78.0% vs. 68.9% for all other diagnoses; logistic regression $P = 2.51 \times 10^{-5}$) as well as the highest burden (median 14.0 vs. 6.9; Wilcoxon $P = 1.12 \times 10^{-8}$). In MDS patients, worse overall survival was associated with EBV presence, even after adjusting for age (Fig. 2d), suggesting potential for EBV as a biomarker for overall survival in MDS. None of the other disease subtypes (AML, MPN, or MDS/MPN) showed an association between EBV and overall survival. We validated our sequence-based inferences of EBV presence/absence and burden using qPCR in a subset of our MDS samples. The results were strongly concordant (Supplementary Fig. 10).

The current standard for risk stratification of MDS patients is the Revised International Prognostic Scoring System (IPSS-R)[28], which uses five risk categories. To explore whether the addition of EBV status could improve survival prediction, we tested for its association with survival within IPSS-R categories, finding that higher EBV status was able to refine risk prediction within the low-risk IPSS-R group. In aggregate, low-risk patients had survival outcomes between very low-risk and intermediate-risk patients (Fig. 2e), as expected, but low-risk patients with EBV reads detected were statistically indistinguishable from intermediate-risk patients, and those without EBV reads were indistinguishable from very low-risk patients (Fig. 2f). We could detect no impact of EBV on survival in the higher risk IPSS categories, likely because any effect of EBV is overwhelmed by the strongly deleterious impact of the risk-defining characteristics (unfavorable blood cell counts and/or cytogenetic abnormalities). To determine more directly whether the impact of EBV on survival is related to the clinical components of the IPSS-R score (hemoglobin, neutrophils, platelets, blasts, and cytogenetics), we tested for association between these components and EBV detection in MDS patients. None of the associations are significant, save that platelet levels are nominally significantly

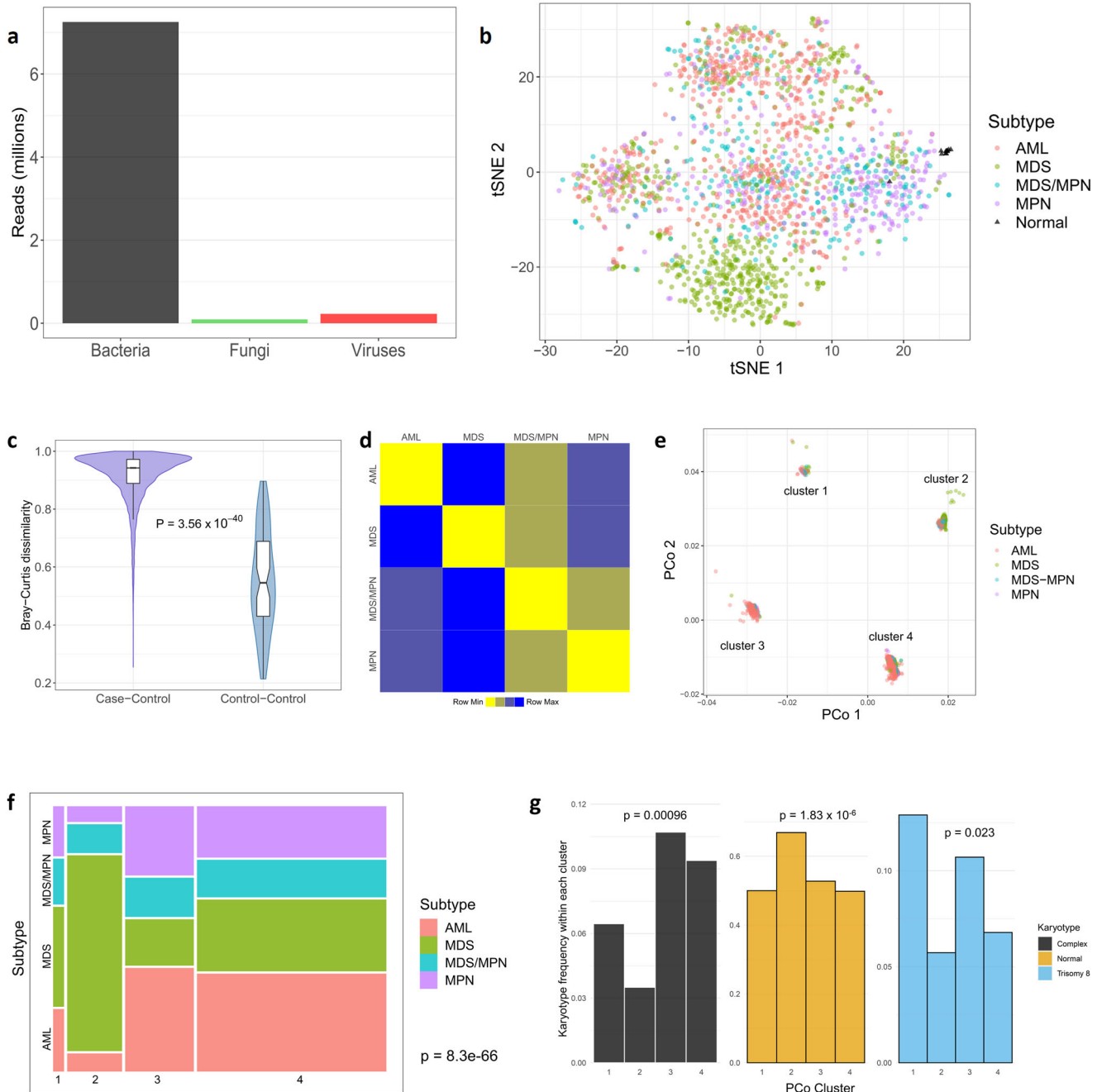

**Fig. 1 Landscape of microbial content in circulation. a** Barplot showing total numbers of reads for each of the three kingdoms. **b** t-SNE plot colored by case/control status (controls shown as black triangles) and disease subtype. **c** Bray–Curtis dissimilarity measures, on the genus level, based for all case-control pairs (left, $n = 22,440$ comparisons) and all pairs of control samples (right, $n = 66$ comparisons). In boxplots, bounds of box indicate first and third quartiles, center line indicates median, and whiskers extend to (first quartile $-1.5 \times$ IQR) and (third quartile $+1.5 \times$ IQR) or extrema, whichever is less extreme (here IQR = interquartile range, i.e. third quartile$-$first quartile). *P*-value computed using two-sided Wilcoxon test. **d** Heatmap representing the average of all Bray–Curtis dissimilarity measures between sample pairs from the indicated groups. Squares are colored according to rank in the row (yellow = most similar, blue = least similar). **e** The first two principal coordinates, on the genus level, colored by disease subtype as in panel (**b**). For clarity, two outliers (an MDS patient and an AML patient) are omitted. **f** Mosaic plot indicating the proportion of the patient cohort in each cluster/subtype pair. The area of each rectangle (colored by subtype) is proportional to the number of patients in the corresponding subtype and cluster. *P*-value from chi-squared test. **g** Barplots indicating proportion of patients, within each principal coordinate cluster, with complex karyotype, normal karyotype, and trisomy 8. *P*-values from two-sided logistic regression-based test. tSNE t-distributed stochastic neighbor embedding, PCo principal coordinate.

lower in EBV-positive individuals (Wilcoxon $P = 0.033$). This overall lack of association suggests that EBV presence is a risk factor that is independent of IPSS-R, which is unsurprising given our observation that EBV refines IPSS-R's low-risk prognosis.

EBV is frequently integrated into the host genome in known EBV-associated tumor types[29], though it is unclear whether these integrations promote malignancy. Given the paired-read nature of the sequencing data, we were able to identify human genome-mapped reads whose mates mapped to the EBV genome, yielding

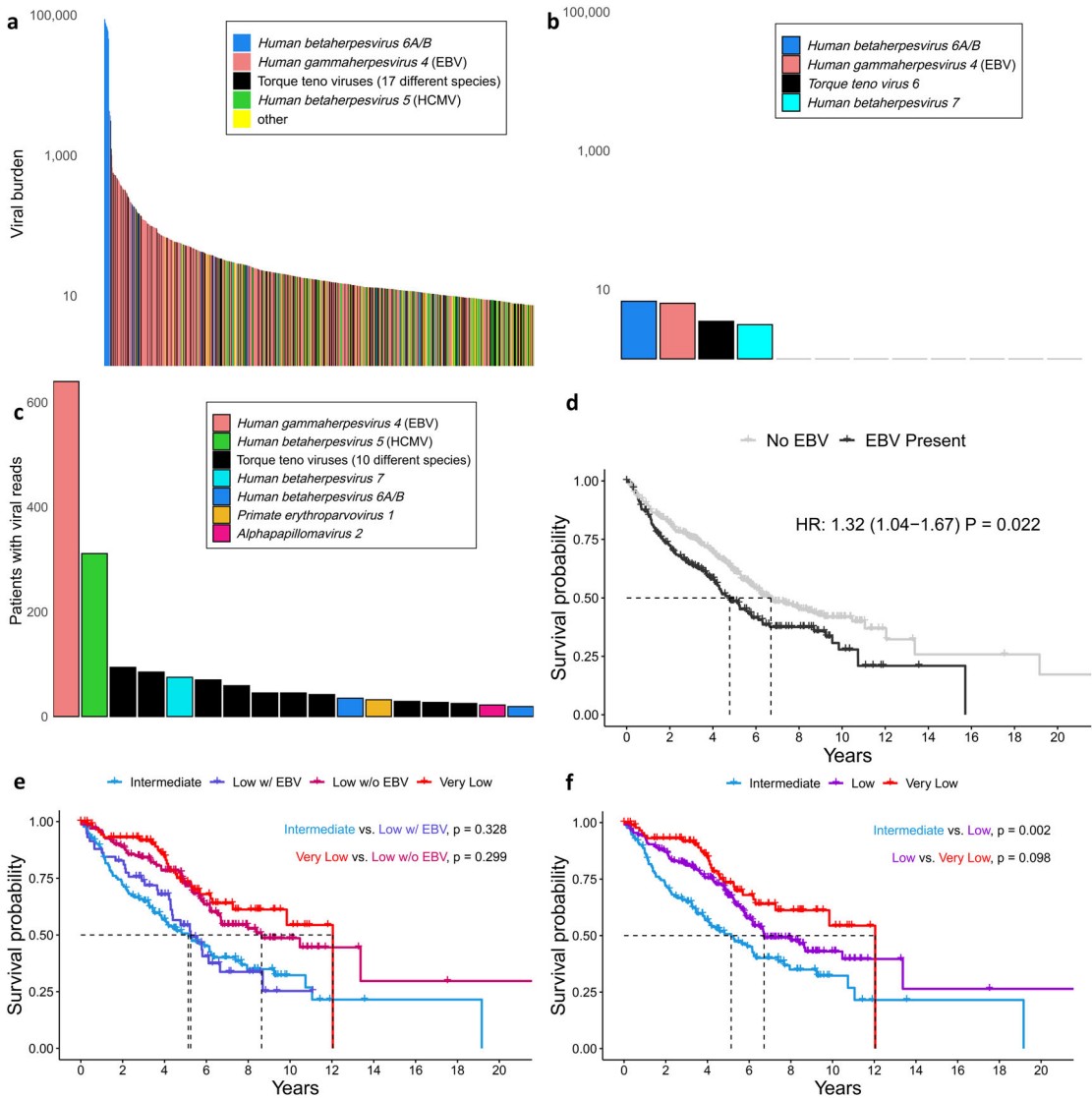

**Fig. 2 Circulating viral content is associated with clinical characteristics. a** Individual species among the top 1/3 of patients with regard to viral burden. **b** All controls are shown with their corresponding detected viruses, on the same (logarithmic) scale as panel **a**, for comparison. Only the leftmost four samples had any detectable viral species. **c** The prevalence of viral species (those found in >1% of cases are shown). **d** Presence of EBV (192 patients with EBV, 448 without) is associated with worse survival in MDS patients (HR and *P*-value are age-adjusted). **E** The Kaplan–Meier curves for intermediate (*n* = 135), low (*n* = 224), and very low (*n* = 92) IPSS-R categories. **f** As in panel **e**, but the low category is stratified by EBV status. Low-risk patients with (*n* = 59) and without (*n* = 165) EBV become statistically indistinguishable from the intermediate-risk and very low-risk categories, respectively. Two-sided *P*-values are computed using the Wald test applied to the Cox proportional hazards model. EBV Epstein-Barr virus; HCMV human cytomegalovirus.

information regarding putative integration sites in our patient cohort. Among the 640 patients with EBV reads, 19 showed evidence of integration, with numbers of putative integration sites ranging from 1 to 15 per patient (Supplementary Data 3). Overall, 47.5% of integration sites were within gene bodies, all intronic. The only recurrently integrated gene was long non-coding RNA LINC00486, which has been identified as a recurrent hepatitis B virus integration site in the liver cancer intrahepatic cholangiocarcinoma[30].

**Fungal prevalence is highest in MDS and is dominated by *Trichosporon asahii*.** As with viruses, fungal reads were found in a higher proportion of MDS patients than those from other disease subtypes (63.3% vs. 56.7%; logistic regression *P* = 0.00586). *Trichosporon asahii* was the most commonly observed fungal species, present in 343 (18.3%) patients. *Trichosporon* infection is commonly reported in patients with acute leukemia[31] and MDS[32], and is a known contributor to mortality

in hematological malignancy patients[33]. However, we found no association between *Trichosporon asahii* presence and overall survival.

**Landscape of the bacteriome in circulation.** The composition of bacteria present in human circulation is known to differ substantially from that in the gut. While gut bacteria are dominated by the phyla Bacteroidetes and Firmicutes[34], studies of the normal blood microbiome consistently demonstrate dominance of Proteobacteria and Actinobacteria[17]. We confirmed the latter composition in our healthy controls (Fig. 3a), with Proteobacteria and Actinobacteria together comprising between 98.8% and 99.9% of all bacterial reads in each sample. Proteobacteria was somewhat more dominant (46.1%–77.4%) than Actinobacteria (21.3%–53.7%). The bacterial landscape in disease cases was substantially different from normal controls (Fig. 3b). Proteobacteria

was generally more dominant in cases (median relative abundance 91.3% vs. 61.1% in controls, Wilcoxon $P = 8 \times 10^{-7}$).

Studies of solid tumors have reported reduced microbial diversity as compared to matched-tissue controls[13,35]. Consistent with these reports, we find reduced α-diversity in cases compared to controls, as measured by the Shannon diversity index (see the "Methods" section), at all taxonomic levels save class (Fig. 3c). For instance, the median α-diversity at the phylum level is 0.33

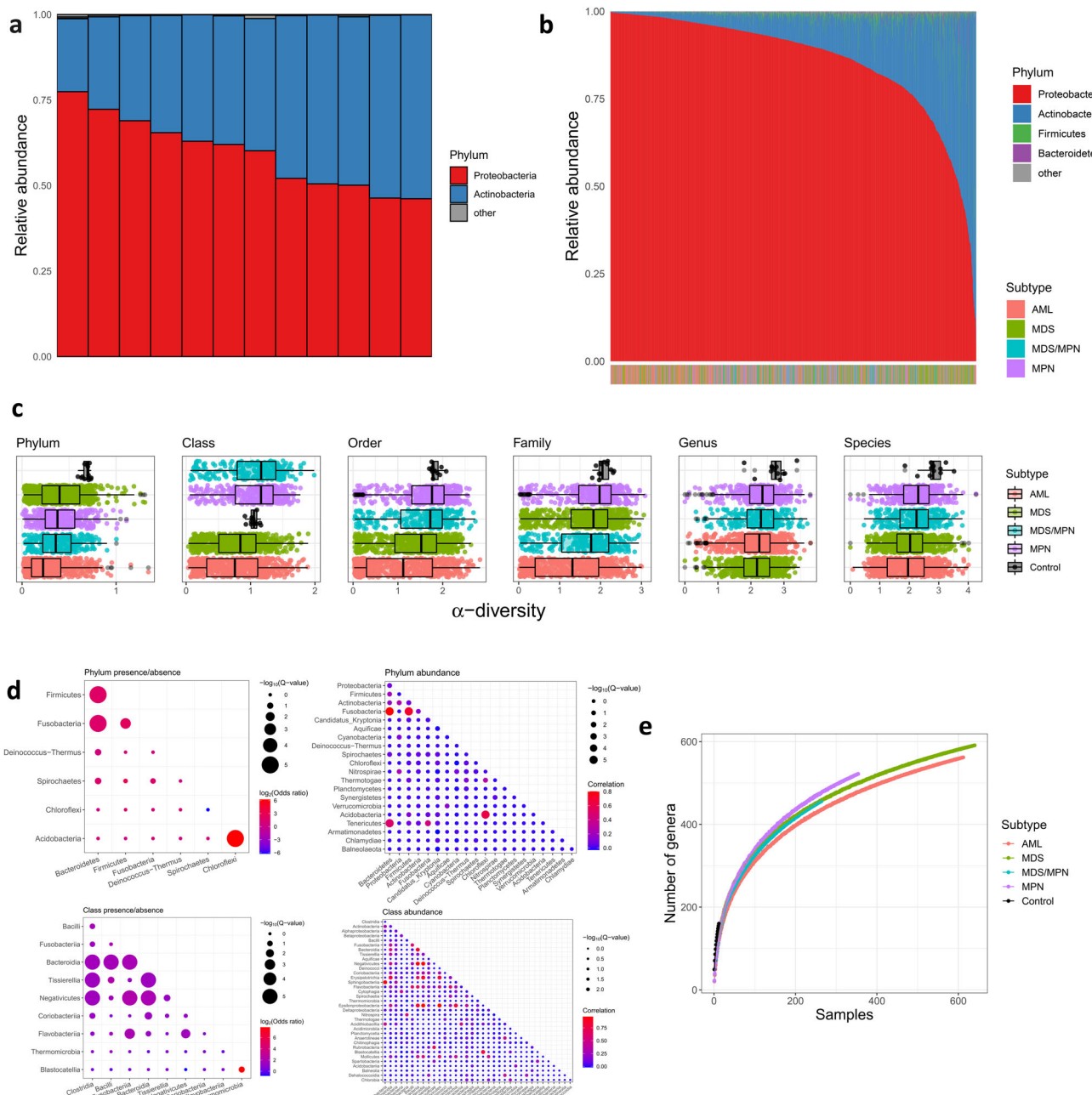

**Fig. 3 The bacterial landscape in the bone marrow/blood of myeloid malignancy patients and controls. a** Relative abundances of phyla are represented by a colored bar for each of the 12 control bone marrow samples. **b** The 1870 colored bars, one for each patient, are ordered left to right by decreasing Proteobacteria relative abundance. The disease subtype of each patient is indicated in the horizontal color bar at the bottom (the enrichment of AML patients among the Proteobacteria-dominant samples is apparent by the color shift at the left side of the bar). **c** α-diversity of each sample within each taxonomic level, stratified by case/control status and disease subtype. Boxplots are ordered top to bottom in decreasing median α-diversity, with sample sizes 12, 612, 640, 264, and 354 for controls, AML, MDS, MDS/MPN, and MPN, respectively. In boxplots, bounds of box indicate first and third quartiles, center line indicates median, and whiskers extend to (first quartile −1.5 × IQR) and (third quartile +1.5 × IQR) or extrema, whichever is less extreme (here IQR = interquartile range, i.e. third quartile–first quartile). **d** Plot showing pairwise concordance/discordance of taxa, at the phylum (top) and class (bottom) levels, both with regard to presence/absence (left) and abundance (right). Sizes of the circles indicate statistical significance, and color indicates strength and direction of association (odds ratio or Pearson correlation). Only taxa with significant (Q < 0.1) concordance/discordance with at least one other taxon are shown. **e** Rarefaction plot showing number of genera as a function of number of patients, stratified by disease subtype. For each patient number n, a random sample of n patients was drawn from each subtype, 500 times. Solid curves represent the mean across the 500 replicates. For control samples, sampling is performed exhaustively (that is, all possible subsets of n individuals are selected for each n = 1,2,...,12).

for cases and 0.70 for controls (Wilcoxon $P = 4.1 \times 10^{-6}$); at the species level it is 2.1 for cases and 2.8 for controls (Wilcoxon $P = 1.3 \times 10^{-4}$).

The reduced diversity in cases raises the question of whether there is competition and cooperation among the various bacterial taxa in patients with myeloid malignancy. To investigate, we tested for correlation/anticorrelation between all pairs of phyla and all pairs of classes. This was assessed both with regard to presence/absence (that is, whether two taxa tend to appear together more or less frequently than would be expected by chance) and burden (whether the burdens are statistically correlated) (Fig. 3d). Assessment of statistical significance, however, is not straightforward in this setting since assumptions of independence are violated. This effect is well known in studies measuring statistically significant mutational concordance/discordance in tumors[36], and renders the use of approaches such as Fisher's exact test prone to false positive discoveries of concordance. As such, we adopted a permutation approach to assess significance of concordance/discordance of taxa (see the "Methods" section). In our analyses, all significant pairs showed positive correlation. Conceivably, this could suggest synergy among different bacterial entities. It could also indicate bacteremia from a common source. For instance the strong positive correlation between Bacteroidetes and Firmicutes, both in terms of presence/absence and burden, could be the result of varying degrees of intestinal barrier permeability (leaky gut) as these are the two most common gut phyla, comprising 90% of microbiota there[37].

### Relationship between bacteriome and clinical characteristics.
The wide range of Actinobacteria–Proteobacteria ratios in our disease cohort led us to inquire whether the ratio corresponded with clinical characteristics. We observed strong association between disease subtype and Proteobacteria relative abundance (age-adjusted ANOVA $P = 7.1 \times 10^{-7}$; also visible in the horizontal bar at bottom of Fig. 3b). In particular, AML had the highest Proteobacteria relative abundance (median 95.0% vs. 89.5% in non-AML cases; logistic regression $P = 2.2 \times 10^{-16}$). AML patients also tended to have the lowest bacterial α-diversity (Fig. 3c) and richness (Fig. 3e).

Given the observed differences in microbial content among disease subtypes (Figs. 1f and 3c, e), we reasoned that microbial taxa might be able to classify patients by subtype. To this end, we constructed machine learning classifiers to identify disease subtype from blood/bone marrow bacterial genus burdens. The genus level was chosen as a compromise between reduced resolution at the higher taxonomic levels and the overabundance of classifying taxa on the species level. For each subtype we developed a binary classifier to distinguish it from all others. Briefly, random forest[38] classifiers were constructed using a randomly selected training subset of the patient cohort. Classifier performance was assessed using the test subset comprised of the remaining samples (see the "Methods" section for details). The classifiers were best able to distinguish AML patients using bacterial content (average area under receiver operating characteristic curve (AUROC) = 0.87, 95% CI 0.84–0.90), though considerable separability was also achieved for MDS (AUROC = 0.84, 95% CI 0.81–0.88) and, to a lesser degree, MDS/MPN (AUROC = 0.75, 95% CI 0.70–0.80) and MPN (AUROC = 0.79, 95% CI 0.75–0.83) (Fig. 4a). We also constructed a pan-subtype classifier that sought to assign subtype to each patient, from among the four (as opposed to distinguishing one vs. the rest), based on genus bacterial burden. We computed AUROC using a method of Hand and Till[39], demonstrating reasonable accuracy for subtype assignment based on microbial content (AUROC = 0.84, 95% CI 0.84–0.84). In general, the performances of our machine learning classifiers provide evidence

that circulating microbial content contains a signal that tracks with disease subtype.

In addition to providing an algorithm to assign classes (disease subtype in our case) based on features (burdens of bacterial genera), a random forest also assigns a measure of variable importance (VI) to each classifying feature. The VI of a feature measures the deterioration of accuracy resulting from obscuring that feature. In our setting, VI can therefore indicate the strength of association of a bacterial genus with each subtype. Among the genera with the highest VI (Fig. 4b, c) are Dermabacter and Kytococcus, both of which have been reported to infect immunocompromised individuals[40,41]. The species Kytococcus schroeteri in particular has been found in multiple patients with myeloid malignancies[42]. Many other genera that contribute to discrimination among subtype (Fig. 4b, c) are known causes of bacteremia, e.g., Staphylococcus, Streptomyces, Rothia, Gordonia, and Pandoraea, among others.

### Microbial differences between AML and MDS are largely driven by myeloblast percentage.
Clinically, AML is distinguished from MDS by the patient's having ≥20% myeloblasts (blasts) in the bone marrow[1]. We therefore next sought to determine whether the microbial characteristics that differ between the two disease subtypes also track with patient blast percentage. Viral read presence (Wilcoxon $P = 0.0013$; Fig. 5a) and burden (Spearman $\rho = -0.104$; $P = 0.00038$; Fig. 5b) were anticorrelated with blast percentage, consistent with higher levels of both observed above in MDS patients. Regarding the bacteriome, blast percentage was correlated with Proteobacteria relative abundance (Spearman $\rho = 0.247$; $P = 1.1 \times 10^{-17}$; Fig. 5c) and anticorrelated with α-diversity at most taxonomic levels (Fig. 5d–i), again consistent with what we observed above with AML patients. Taken together, these data demonstrate an association between blast percentage and microbial characteristics and suggest that the microbial differences we observe among different disease subtypes are at least partially driven by blast percentage.

To investigate this association further, we next modified our machine learning classifier to distinguish between patients with blast percentages above and below varying threshold levels. That is, we built multiple machine learning classifiers, one for each of varying blast percentage thresholds $bp$, to distinguish patients with blast percentage ≥$bp$ from those with blast percentage <$bp$ based solely on their bacterial content. Each classifier's performance was then assessed by its AUROC. As shown in Fig. 5j, the optimal threshold is 18% (AUROC 0.88, 95% CI 0.86–0.91), i.e. the bacteriome most accurately distinguishes between patients above and below 18% blast percentage, intriguingly quite close to the somewhat arbitrary 20% threshold used clinically.

### Relationships between bacteriome and host mutations.
Recent work has shown that AML may be spurred by gut microbiota that enter circulation via host mutation-induced intestinal permeability[43]. To investigate possible impact of host mutation on the circulating microbiome, we tested the relationship between microbial characteristics and mutations in genes commonly mutated in myeloid malignancy (ASXL1, CALR, CBL, CEBPA, DNMT3A, EZH2, FLT3 point mutation, FLT3 internal tandem duplication (FLT3-ITD), IDH1, IDH2, JAK2, KRAS, NPM1, NRAS, RUNX1, SETBP1, SF3B1, SRSF2, STAG2, TET2, TP53, U2AF1, WT1, and ZRSR2). Specifically we tested for differences, between mutant and wild-type patients for each gene, in overall bacterial burden, genus α-diversity, Proteobacteria relative abundance, and Firmicutes relative abundance + Bacteroidetes relative abundance (this last characteristic was used as a proxy for potential gut permeability, since Firmicutes and Bacteroidetes are the two dominant phyla in gut microflora). This analysis entailed

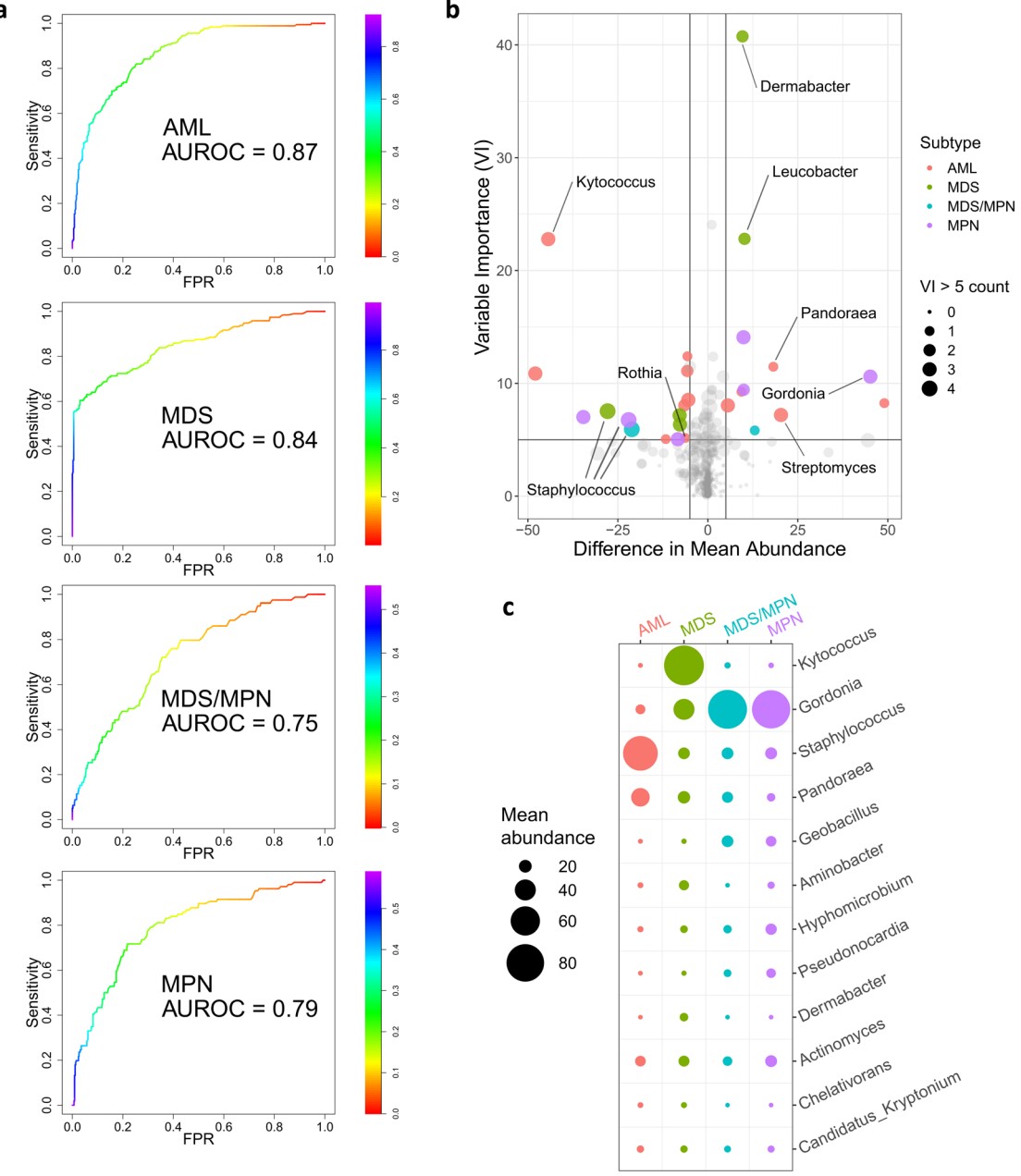

**Fig. 4 Bacterial composition differs among disease subtypes. a** ROC curves showing, for each disease subtype, the performance on the test set (randomly selected 30% of samples) of binary random forest classifier trained on the training set (remaining 70%). The AUROC values shown are averaged across 1000 random 70%/30% splits. The random forest generates a probability of a sample having the disease subtype in question. The color bar indicates varying thresholds of this probability. **b** Volcano plot showing enrichment/depletion of bacterial genera in specific disease subtypes. Here horizontal axis indicates differences in mean abundance (subtype of interest—all others), and variable importance is shown on the vertical axis. Point size indicates number of subtypes (0 = smallest, 4 = largest) for which the corresponding genus has variable importance >5. Points with mean abundance difference >5 and variable importance >5 are colored by corresponding subtype. Points of interest are labeled with their corresponding genera. (VI variable importance). **c** Mean abundances, in each subtype, of the genera that are among the top five in variable importance for at least one of the subtypes. Circle size indicates the average abundance in the corresponding subtype. AUROC area under receiver operating characteristic curve; FPR false positive rate.

96 statistical tests, of which three achieved Bonferroni-corrected statistical significance ($P < 0.05/96$): *DNMT3A* mutations were associated with lower genus α-diversity (logistic regression $P = 0.00027$; Fig. 6a), while *FLT3* and *NPM1* mutations were associated with higher Proteobacteria levels (logistic regression $P = 0.00052$ and $0.00030$, respectively; Fig. 6b, c). Both *FLT3* and *NPM1* are among the most frequently mutated genes in AML[44] but are rarely mutated in MDS[45,46]. It is therefore possible that

the observed associations with Proteobacteria levels are simply a consequence of higher Proteobacteria levels in AML.

Among MPN patients, the *JAK2*-mutant cases are considered a distinct clinical entity. We compared the microbial composition of *JAK2*-mutant ($n = 162$) with that of *JAK2*-wild type ($n = 181$) MPN patients but found no statistical differences in the four microbial characteristics tested above. We next built a machine-learning classifier to distinguish *JAK2*-mutant from *JAK2*-wild

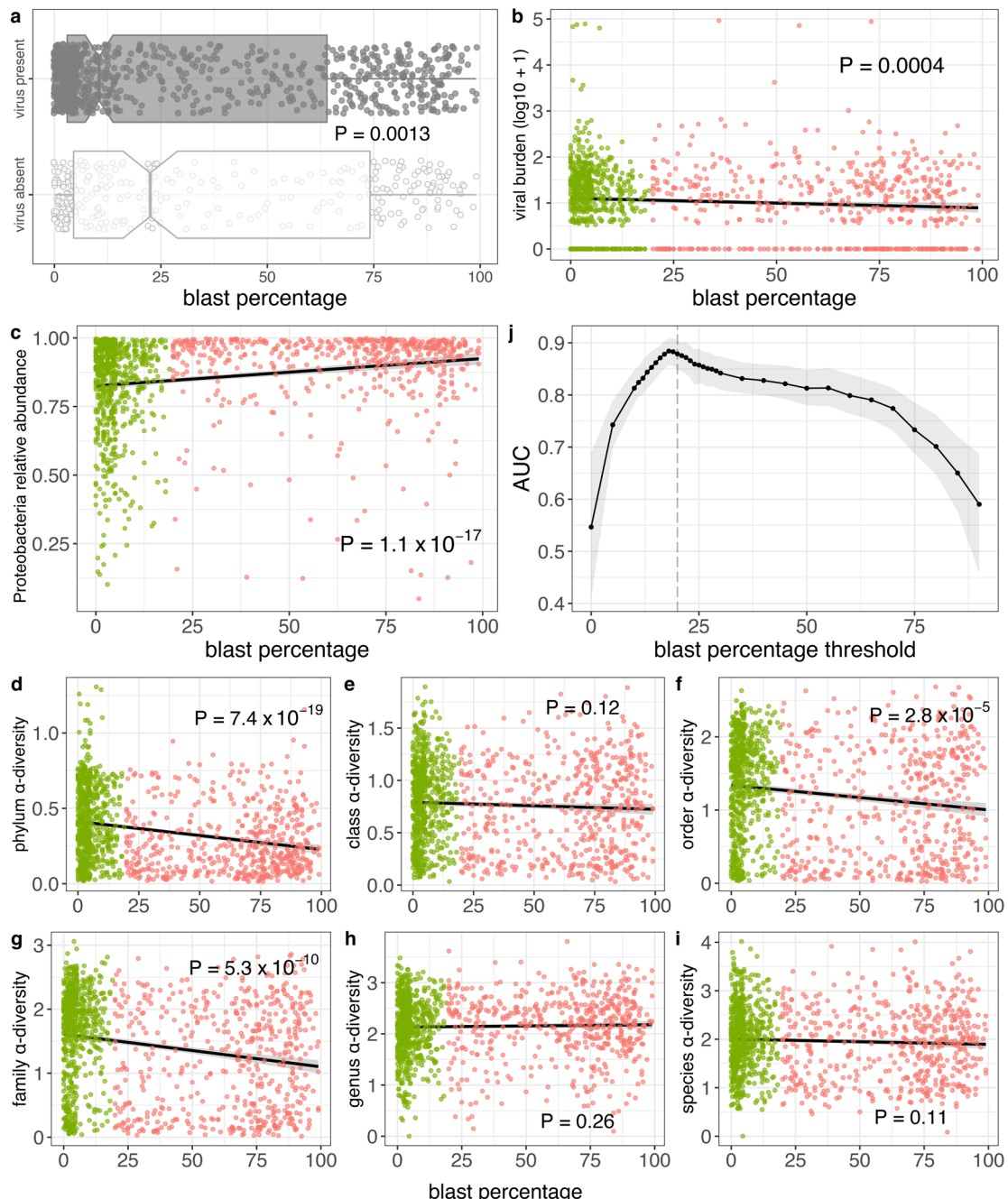

**Fig. 5 Associations between myeloblast percentage and microbial characteristics. a** Viral read presence is associated with lower blast percentage. Here 868 patients have virus present and 296 patients have virus absent. *P*-value from two-sided Wilcoxon test. In boxplots, bounds of box indicate first and third quartiles, center line indicates median, and whiskers extend to (first quartile −1.5 × IQR) and (third quartile +1.5 × IQR) or extrema, whichever is less extreme (here IQR = interquartile range, i.e. third quartile−first quartile). **b** Viral burden are both associated with lower blast percentage. **c** Proteobacteria relative abundance is positively correlated with blast percentage. **d–i** For most taxonomic levels, α-diversity is negatively correlated with blast percentage. **j** AUROCs for random forest classification of patients ($n = 1164$) above/below various blast percentage thresholds, with mean over 1000 independent training/test splits indicated in black and 95% confidence intervals indicated with gray shading. In panels **b–i**, green indicates MDS patients ($n = 638$), salmon indicates AML patients ($n = 526$), shading indicates the 95% confidence interval, and *P*-values are from two-sided Spearman correlation test.

type MPN patients from microbial content, much in the same manner as above for the subtype classifiers. The performance of the *JAK2* classifier was not as strong, but the AUROC (0.63, 95% CI 0.54–0.71) was significantly above 0.5 (Fig. 6d). This result suggests some relationship between *JAK2* mutation and the circulating microbiome, though further study is required to validate the relationship and elucidate its nature.

## Discussion
Here we have reported results of a large, high-resolution survey of microbial content in the blood and bone marrow of myeloid malignancy patients. We have cataloged bacterial, fungal, and viral content in circulation for 1870 disease cases and 12 healthy controls, all processed and sequenced at the same center. Our overarching aims were to investigate the relationships between

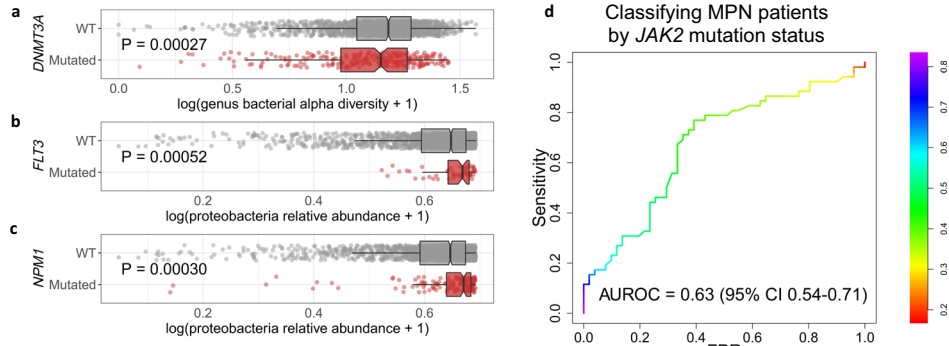

**Fig. 6 Associations between gene mutations and microbial characteristics. a** Genus α-diversity stratified by *DNMT3A* mutation status (1580 WT, 259 mutated). **b, c** Proteobacteria relative abundance stratified by *FLT3* (1804 WT, 56 mutated) and *NPM1* (1698 WT, 167 mutated) mutation status. In boxplots, bounds of box indicate first and third quartiles, center line indicates median, and whiskers extend to (first quartile −1.5 × IQR) and (third quartile +1.5 × IQR) or extrema, whichever is less extreme (here IQR = interquartile range, i.e. third quartile–first quartile). *P*-values are from a two-sided logistic regression-based test. **d** ROC curve for random forest algorithm to classify MPN patients by *JAK2* mutation status from microbial content. The random forest generates a probability of a sample having a *JAK2* mutation. The color bar indicates varying thresholds of this probability for calling the sample as having the mutation. WT wild type, AUROC area under receiver operating characteristic curve, CI confidence interval, FPR false positive rate.

disease subtypes/patient outcome and the circulating microbiome in myeloid malignancies.

Interestingly, we did not observe any strong differences in microbial content between patient samples taken from peripheral blood and those taken from bone marrow, suggesting that the microbes and/or their DNA are transported freely into and out of the marrow through the bone vessels and sinusoids. However, it is unknown if the species detected here are living or active, and, if living, whether they are extracellular or have entered human cells. Even active microbes may be in circulation only transiently, likely translocating from the gut, skin, or mouth[47,48]. Strong effects of age and sex on microbial content were also not detected.

Our study provided strong evidence for substantial dysbiosis in the circulating microbiome of myeloid malignancy patients. The patients had significant shifts in dominant bacterial phyla as compared to healthy controls, and a reduction in α-diversity. This dysbiosis may partially be explained by intestinal permeability in some patients. Intestinal permeability is present in myeloid malignancy patients even before treatment[49], and has been recently implicated in leukemia development in mice[43,50]. All samples analyzed here were taken at diagnosis, and therefore microbial content would not be influenced by therapy.

In 1954 Ludwig Gross hypothesized, based on mouse experiments, a viral cause for human leukemia[51]. Although the hypothesis has not been validated for most leukemias (with some exceptions), recent work has shown associations between viral content and leukemia patient outcomes[52,53]. In our cohort, cases had a much higher viral burden than controls, largely from herpesviruses EBV and HCMV. Viruses and fungi were most prevalent in MDS patients, and our analysis revealed association of EBV with overall survival in MDS. Additional research is required to determine whether the high levels of viral burden among disease cases play a causal role or are instead a consequence of the immunosuppressive effects of the disease.

On the bacterial level, the landscape differed significantly among the four disease subtypes, with multiple taxa showing significant differences among the subtypes. AML patients had significantly higher bacterial burden but lower diversity, perhaps reflecting the dominance of Proteobacteria in AML patients. These observations motivated us to develop machine learning classifiers to predict subtype from microbial content. The classifiers showed the ability to separate each subtype from the other three, demonstrating promise for further refinement of this approach. Our results here are analogous to recent microbe-based

classifiers that were able to distinguish between different stages of some solid tumors[16]. Similarly, bacterial taxa have shown differing prevalence among breast cancer subtypes[15]. In our setting, the differences in bacterial signature among subtypes could be the result of varying degrees of immunocompromise among the four entities, leading to differential ability to combat bacteremia. Hematopoietic cell composition in circulation could also affect microbial content. AML patients have a higher prevalence of neutropenia, and absolute neutrophil counts (ANC) vary greatly by disease subtype. There is the possibility that the observed associations could be the result of confounding factors. For instance, differences in antibiotic use among the different subtypes would likely produce subtype-specific microbial signatures. We do not have data on patient medications, but it is important to note that all samples analyzed in our study were taken at diagnosis, and any antibiotics differentially prescribed because of subtype would not yet have affected these signatures.

A natural extension of the work presented here would be to query the circulating microbiomes of patients with MDS precursor conditions such as clonal hematopoiesis of indeterminate potential (CHIP) and clonal cytopenia of undetermined significance (CCUS)[54]. Given the differences in microbial characteristics that we observed among different disease subtypes, risk categories, and gene mutation statuses, one might hypothesize that individuals with CHIP or CCUS would show microbial signatures that are intermediary between normal and MDS (particularly low-risk MDS) signatures, and that the CHIP/CCUS-defining mutation(s) may track with microbial characteristics. Such studies would shed further light on the implications and potential mechanisms of the associations reported in the current work.

The strengths of our study include a large cohort of disease cases, all processed and sequenced at the same facility along with 12 healthy controls. We used shotgun metagenomic sequencing, which has advantages over 16S sequencing, including better taxonomic resolution, with typically higher revealed diversity[55]. It also enables ascertainment of viral and fungal content along with bacterial content. We took a very aggressive approach to data filtering, taking care to remove all reads that were likely artifactual results of contamination or mis-mapping of human reads. The fact that our filtering approach removed more than 99% of the data serves as a warning that it is critically important to take great care in microbiome studies (particularly those analyzing low-microbial biomass samples).

Our study also had weaknesses. It did not include an independent validation cohort owing to the considerable expense in performing shotgun sequencing in a large sample of myeloid malignancy patients with disparate subtypes. It should be noted, however, that our disease subtype classifier was validated to some degree by building it on a subset of the cohort and testing it on the remaining samples. Another issue is that our data was derived from samples processed for human DNA analysis. Ideally, we would have included technical controls[56] in our study, to account for artifacts specific to our experimental protocols. Since the samples were originally collected in clinical practice for mutation profiling, such controls were not available. Nonetheless, our computational approach was conservative, omitting a large list of known reagent and kit contaminants along with other artifacts. Finally, we acknowledge that our study has a relatively small number of healthy controls as compared to cases. Overall, we felt that the importance of using controls well-matched to cases in terms of tissue source and processing/sequencing site outweighed the statistical power gained by seeking other sources of data. Using externally-processed controls could lead to batch effects and artifactual results. As our results show, however, we were still well-powered to uncover a number of differences in microbial composition between the two groups.

In conclusion, this report serves as an initial baseline for future studies of the microbiome in circulation of myeloid malignancy patients. As growing evidence emerges that response to treatment may be influenced through gut microbiome perturbations, the results reported here may shed light on the potential for analogous manipulation of the blood microbiome to favorably impact patient outcomes[57]. Future work to replicate our findings in separate cohorts is crucial, and functional experiments are needed to determine whether and how microbes influence the course of disease. Such experiments will shed light on the potential of bacteria, fungi, and viruses to serve as biomarkers in myeloid malignancies, and may suggest treatment options for a subset of patients.

## Methods

**Disease subtype assignment, sample acquisition and processing, and whole-genome sequencing**. For all 1870 cases, diagnosis and disease subtypes were assigned using cytomorphology, immunophenotyping, cytogenetics, and molecular genetics following World Health Organization (WHO) guidelines. All patients gave their written informed consent for scientific evaluations. The study has been approved by the Internal Review Board of the Munich Leukemia Laboratory as well as by the ethics committee of the Bavarian physicians´ chamber and adhered to the tenets of the Declaration of Helsinki. Additionally, 12 bone marrow samples from healthy donors were included as controls. All donors gave their written informed consent for scientific evaluations. Complete cytogenetic data according to ISCN nomenclature[58] is available for all patients by request at Munich Leukemia Laboratory.

For whole genome sequencing (WGS), peripheral blood or bone marrow aspirates were processed using the TruSeq DNA PCR-free library prep kit and 150 bp paired-end sequences were generated on a NovaSeq 6000 or HiSeqX instrument (Illumina, San Diego, CA). Fastq generation and read alignment to the human reference genome were performed using Illumina's BaseSpace platform (whole genome sequencing app 5.0.0).

**Identification and quantification of microbial reads**. Whole-genome bam files were converted back to fastq files using the GATK4[59] SamToFastq tool. The resulting fastq files were then aligned to the hg19 build of the human genome using bwa mem[60], yielding bam files that served as input into PathSeq[19], distributed as part of GATK 4.0.6.0. Briefly, PathSeq first removes all human genome-aligned reads, then aligns those remaining to an NCBI database of known microbial reference genomes. Default options were used with parameters --min-clipped-read-length 70 and --is-host-aligned true. Required reference files (microbe-fasta, microbe-bwa-image, and taxonomy-file) were downloaded as part of the GATK resource bundle. The output of PathSeq provides, for each taxon and patient, counts of reads that could be unambiguously assigned to that taxon. After filtering steps (see below), burden of taxon $i$ in individual $j$ was quantified as

$$6.4 \text{ billion} \times \frac{\text{number of reads aligning unambiguously to taxon } i \text{ in individual } j}{\text{number of reads aligning to the human genome in individual } j} \quad (1)$$

The rationale for this measure is that it estimates the number of bases of the taxon DNA present per human cell, since there are ~6.4 billion bases of human sequence per human cell.

**Quality filtering**. Reads deemed unambiguously aligned by PathSeq were subjected to two filtering steps. First, we curated a list of genera and species that were reported in the literature as being problematic for various reasons, including: (i) contamination in commercially available kits and reagents; (ii) common low-read levels across tumor types; (iii) anticorrelation between measured abundance and analyte concentration; (iv) high frequency in negative blanks; and (v) artifactual human sequence within species reference genomes. Second, alignments of species were manually examined for their locations in the microbe genome. Species that had reads only aligning to focal regions of their genome were flagged as problematic. We removed all reads unambiguously aligned to these problematic taxa and propagated the removal up the taxonomic tree. For example, if a problematic genus had 20 unambiguously aligned reads in that patient, then 20 reads would also be removed from that genus' parent family, order, class, and phylum. Furthermore, all daughter species of the genus and their reads would be removed from further analysis.

**Computing microbial landscape characteristics**. Let $r_{ij}$ denote the number of unambiguous reads from taxon $i$ in individual $j$. Then the relative abundance for the taxon in that individual is computed as $r_{ij}/T_j$, where $T_j$ denotes the total number of reads in that individual that map unambiguously to a taxon at the same taxonomic level as taxon $i$. The t-SNE coordinates were generated from the matrix giving the relative abundance of each genus for each sample. A series of pre-processing steps was first implemented as suggested by Kobak et al.[61], then FIt-SNE (FFT-accelerated Interpolation-based t-SNE)[62], a variant of t-SNE algorithm, was used to generate the coordinates.

Bray–Curtis dissimilarity statistics were calculated for each pair of samples using burden. For each pair of samples $x$ and $y$, Bray–Curtis dissimilarity is calculated across $n$ genera as

$$\frac{\sum_i^n |x_i - y_i|}{\sum_i^n (x_i + y_i)} \quad (2)$$

where $x_i$ and $y_i$ denote the burden of genus $i$ in sample $x$ and $y$, respectively. Then the dissimilarity measures were used to generate principal coordinates of this neighborhood matrix using the pcnm function in the vegan package (version 2.5-6).

The α-diversity for a taxonomic level was calculated for patient $s$ as

$$-\sum_i^t p_i \ln p_i \quad (3)$$

where $p_i$ is the proportion of unambiguous reads at the taxonomic level that map to taxon $i$ within patient $s$, and $t$ is the total number of taxa observed in patient $s$ within the taxonomic level.

**Statistical analyses**. All statistical analyses were performed using R version 4.0.3. Reported $P$-values are two-sided. To assess the significance of concordance/discordance between all pairs of taxa, the presence/absence of all $n$ taxa within the same taxonomic level was represented as an $n \times 1870$ matrix, where the rows represent the $n$ taxa, the columns represent the 1870 patients, and entry $(i, j)$ is 1 if patient $j$ has detected presence of taxon $i$ and 0 otherwise. The odds ratio for each pair of taxa is computed in the observed data. To determine the statistical significance of each odds ratio, we first repeatedly permuted the data matrix in a manner that keeps the row and column sums (the total number of patients harboring each taxon, and the total number of observed taxa within each patient, respectively) constant. In this way, we preserve taxonomic richness for each patient and overall frequency of each taxon. Permutations were performed using the permatfull function in the vegan package, using seed 2021 and parameters fixedmar = "both", shuffle = "both", mtype = "prab", and times = 100, and odds ratios computed for all pairs in each permutation. The $P$-values corresponding to each odds ratio $x$ in the observed data is computed as the proportion of permuted odds ratios as or more extreme as $x$. The $Q$-values were computed from each observed $P$-value by dividing the average number of permuted $P$-values lower than or equal to the observed $P$-value (false discoveries) by the total number of observed $P$-values lower than or equal to the observed $P$-value (discoveries).

Logistic regression $P$-values were computed using the R command anova(mdl, test = "Chisq"), where "mdl" is the fitted logistic regression model.

Survival analysis was performed using the R packages survival (3.2–7) and survminer (0.4.8). Age-adjusted hazard ratios and corresponding confidence intervals and $P$-values were obtained by fitting Cox proportional hazards regression models using the coxph function.

**qPCR validation of EBV**. The total genomic DNA isolated from the patient-derived mononuclear cells were used for the EBV detection using real time quantitative polymerase chain reaction (RT-qPCR)[63,64]. Briefly, the primers for detection of EBV were designed to target the BamHI-W sequence in EBV using following pairs of forward and reverse primers: 5′-CCAGACGAGTCCGTAGAAGG-3′ and 5′-AGCCTAATCCCAC CCAGACT-3′, respectively.

The 96 ng of each of DNA samples were added in per PCR reaction for Sso Fast EvaGreen Super mix (Bio Rad) and subjected to 40 amplification cycles in C1000 Touch™ thermal cycler (Bio Rad Inc) interfaced with CFX 96 system manager software. The TAL57 region of the Human Beta-Globin (HBB) gene was used for control using 5′-TAGCAACCTCAAACAGACACCA-3′ and 5′-CAGCCTAAGGGTGGGAAAAT-3′ forward and reverse primers respectively. The $C_t$ values on RT-qPCR were collected by the CFX Manager Software (Bio Rad Inc) and the relative copies/μg DNA was calculated by $\Delta C_t$ method.

**Calling EBV integration sites**. To identify putative EBV integration sites, all read pairs with one end mapping to the human genome and the other mapping to the EBV genome were flagged. The mapped position in the human genome was reported as the putative EBV integration site, after confirming that the human-mapped read did not map to the EBV genome.

**Adjusting viral prevalence for genome length**. For a given viral species and a given patient, let $r$ denote the number of reads mapping to that species in the patient sample, and let $s$ denote the species' reference genome length. If the species instead had a reduced genome of length only 2442 (the length of the smallest reference genome among all viruses we observed in our study), for each of the $r$ reads, the probability of its mapping to a specific reduced genome, 442-base region is $\sim 2442/s$, so the probability that all $r$ reads do not map to the region (i.e. the probability that the virus is not observed in the patient) is $(1-2442/s)^r$, giving the probability of observing the virus in the patient as $1-(1-2442/s)^r$. If we now consider all $n$ patients, with number of reads mapping to the species' actual genome denoted $r_i$ ($i = 1,…,n$), the expected number of patients in which at least one read maps to the reduced genome (i.e. the expected prevalence, adjusted for genome length) is therefore

$$\sum_{i=1}^{n}\left(1-\left(1-\frac{2442}{s}\right)^{r_i}\right) \quad (4)$$

**Disease subtype/*JAK2*-mutant classifiers**. A classifier to predict one disease subtype against all others was built by training a random forest using the randomForest R package (4.6–14) with default parameters. The model was trained on the bacterial genus burdens in 70% of the samples, and performance was assessed on the remaining 30%. ROC curves were built for the classification model using R package PRROC (1.3.1). Here a sample is deemed to have the disease subtype in question if the proportion of trees classifying it as such exceeds a threshold. The ROC curves are generated by assessing sensitivity and false positivity at each value of the varying threshold. AUROC was used to evaluate performance on the test set of binary classifiers using the ROCR R package (v1.0-11), and of the multiclass classifier using the pROC R package (v.1.18.0), the latter implementing the method from Hand and Till[39]. To test whether the model assessments were robust, the process of training random forest models on 70% of samples and assessing performance on 30% (keeping the relative proportions of disease subtypes equivalent in training and test sets) was repeated 1000 times for each subtype. 95% confidence intervals around the mean of the AUROCs from the 1000 models were determined as the 2.5th and 97.5th percentiles from the 1000 AUROCs calculated. The same procedure was used to build and validate the *JAK2*-mutant classifier.

**Reporting summary**. Further information on research design is available in the Nature Research Reporting Summary linked to this article.

## Data availability

Bam files generated in this study containing all non-human reads have been deposited in the NCBI's database under accession code PRJNA746290. Human genome build hg19 can be downloaded from https://hgdownload.soe.ucsc.edu/goldenPath/hg19/bigZips/hg19.fa.gz.

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

## Acknowledgements

This work was supported by US National Institutes of Health grants R01LM013067, R01CA217992 and R21CA249138 (to T.L.), R01CA257544 (to B.K.J. and J.P.M.), and the Torsten Haferlach Leukamiediagnostik Stiftung.

## Author contributions

J. Woerner, Yidi Huang, J.M.H.S., J. Wang, Yimin Huang, D.S., M.A., W.X., V.T., and T.L. performed data analysis. S.H. and T.H. conducted the whole-genome sequencing. S.H., C.G., T.H. and J.P.M. advised on the clinical aspects of the work. D.J. and B.K.J. performed PCR validation experiments. M.K. advised on the computational aspects of the work. T.L. supervised the work. T.L. and J. Woerner wrote the paper.

## Competing interests

The authors declare no competing interests.
