## [Peer Review File · Nature Communications]

Circulating microbial content in myeloid malignancy patients
has diagnosis-associated characteristics and prognostic
potentialREVIEWER COMMENTS

Reviewer #1 (Remarks to the Author):

In this study, the authors studied 1,870 unique patient samples (mostly bone marrow) that had been sent to the Munich Leukemia Laboratory between 2005 and 2017. Diagnoses including MDS, AML, MPN, and MDS/MPN. They also processed bone marrow from 12 healthy donors. These samples were initially sequenced to profile somatic human mutations. Sequencing was performed using 150bp paired-end reads. They filtered the reads extensively looking for reads mapping to microbes, with most reads mapping to bacteria (Fig. 1A). Table 1 shows the patient characteristics including karyotypes. Figure 1 shows some grouping by disease type with distinction relative to healthy controls. Figure 2 shows data from viral reads focusing on how finding reads from EBV is associated with worse prognosis. Figure 3 shows the predominance of bacterial reads mapping to Proteobacteria. Figure 4 shows the receiver operating curves for predicting malignancy type based on microbial taxa.

The authors are to be commended for their exhaustive efforts to characterize the microbiome of bone marrow/blood samples from an extremely robust cohort of metagenomic data of myeloid malignancy patients. They have gone to significant efforts to control for DNA contamination and have done excellent downstream analytics. They have found that there are differences amongst the microbial content in the bone marrow/blood in various myeloid malignancies, that the presence of reads mapping to EBV adds to the prognostic classification in low-intermediate risk MDS, and that Proteobacteria reads are higher amongst AML patients relative to the others. These findings are novel and could spur new research avenues amongst hematologic malignancy patients. Statistical analyses seem appropriate.

That said, there are some major issues with the manuscript:

1. Comparing 1,870 samples with 12 controls is highly problematic, particularly given the known diversity of the microbiome amongst healthy persons. All comparisons to healthy controls should be done with significant caution. Having a more robust group of healthy controls would strengthen such comparisons.
2. The samples were analyzed over a 12 year period which raises concern for introduction of various processing heterogeneity into the data. It is known that processing effects can have a major impact on microbiome studies so the authors should do an analysis looking at whether their microbial content changed in a temporal fashion.
3. The samples were not processed initially to maximize microbial DNA but rather were focused on human DNA. Thus, the processing itself could have led to significant artifact in terms of what microbial DNA remained. This cannot be undone given the nature of the study but should be cited as a limitation of the study.
4. The authors should confirm some of their findings with targeted quantitative PCR. Given concerns for contamination and other mapping artifacts, targeted qPCR confirmation would be quite helpful. There are well established protocols for this for herpes viruses as well as various bacterial genera. Their finding that EBV presence can be used as a prognostic marker in low-intermediate risk MDS would particularly benefit from qPCR analysis.

Specific comments:

1. Please check the y-axes on panel 1G. I do not see how the proportions add up to 1.
2. Although the authors claim their data suggests the ability to provide diagnostic criteria, in reality the AUCs are not strong. It is not realistic to think that a microbiome profile would be used to diagnose (and hence design treatments) for particularly myeloid malignancies. It is interesting that there are discriminative properties of their microbiome analyses but not to the point of using such data to diagnose.
3. For Figure 2, HHV-6 is well known to integrate into the germ-line of about 1% of humans leading to high-level viremia (see PMID 32044155). Probably this explains the high levels observed in some patients in Fig. 2A.
4. It is not particularly surprising that bone marrow and blood samples from myeloid malignancy

patients would have high levels of EBV and CMV (to a lesser degree). These viruses are near universally present in humans and are latently present in hematologic cells with reactivation occurring during periods of immunosuppression. Although the authors touch on this in the discussion, these findings could more clearly be presented in the setting of what is known about these viruses.

5. The y-axis of Fig. 2C read “Number of affected patients” whereas actually this is the number of patients samples in which viral reads are detected. There is no clinical data that these patients were “affected”.

6. Similarly, the authors write that “differences in viral infection...” on page 4. Infection implies some type of clinical issue. These viruses all establish latency in human cells so finding DNA does not imply infection. Similarly, the finding of reads mapping to *Trichosporon asahii* does not imply infection (page 5).

7. The rarefaction curve (Fig. 3E) is not discussed in the manuscript that I could find.

8. On page 6, the authors state that *Dermabacter* and *Kytococcus* are opportunistic pathogens frequently affecting immunocompromised individuals. In reality, these organisms are exceedingly rare causes of infections.

9. In the discussion the authors write that they did not observe significant differences between blood marrow and blood samples, but these data are not presented. It is important to do so because the samples are considered as a group (and typically the authors write about bone marrow samples when including both samples types).

Reviewer #2 (Remarks to the Author):

The authors explored the bone and blood microbiome of a large number of 1870 patients in order to identify microbial associations with cancer and the potential to predict four tumor subtypes of myeloid malignancy diseases.

The work presents some interesting microbial candidates to discriminate tumor types and shows a diagnostic potential of the microbial composition. But individual microbial taxons could not be observed to work as a single strong marker that shows significant differences between disease types. Also the predictive power of the microbial total community has some diagnostics limits as healthy controls were not included in the classification analysis, and hence it's a discrimination between disease types alone and not between healthy and disease.

The weakest point of this study is clearly the small number of 12 controls versus 1870 disease cases in the taxonomic analysis and a complete absence of healthy controls in the diagnostic analysis.

Controls are also absent in PCoA plot Fig. 1e, but present in Fig. 1b.

Why are healthy controls sometimes included and sometimes not?

Another limiting factor is of course the amount of microbial sequences that could be extracted from bone and blood samples. Per sample counts of <4000 reads for bacteria and 120 for viruses are very small for reliable taxonomic identification.

To get a better picture of the microbial sequencing depth, it would be helpful to report not only the mean, but also the std, min, and max numbers of microbial related reads across all samples.

And also what are the total numbers of raw read counts or base pairs per sample? Do the 12 control samples have a total number of raw reads comparable with disease samples (similar sequencing depth)?

Taxonomic identification: While the use of PathSeq might work to count taxa related reads, the described need to exclude genomes having unbalanced or peaked converge regions could be avoided by using a taxonomic identifier like MetaPhlan that does not simply count reads, instead claims a taxon to be present when hundreds of marker genes are present, hence forcing a more broad genome coverage. Or, alternatively, a kind of measure like median coverage across all bases of a genome could be applied instead of error prone manual examination.

Figure 2a,b: Control samples may have a lower viral abundance, but comparing only the top highest 100 samples of 1870 disease samples with all controls is not correct. If 4 of 12 control samples (33%) carry viruses, Fig 2a has to show also 33% or >600 samples of case samples to have a fair visual comparison.

Figure 2c: Virus prevalence might be affected by the virus genome length. Working close to the detection limit (few virus related reads per sample), large virus genomes might be easier identified than small sized genomes. Is the genome size somehow considered or are there any corrections done?

Identifying virus integration sites: Yes, paired reads with one end mapping to the human genome and the other to virus genome, might be a hint of host integrated sequences. However, it's important to double check secondary read alignments in order to exclude random mapping on identical regions present in host and virus. This could be done by mapping these host-virus reads only to the human genome to confirm that only one read of a pair can be aligned to the host genome. The same mapping can be done using only the virus genome.

Alternatively, instead of mapping against the hg19 human reference genome, the sample specific host genome could be reconstructed. Having a 100X human genome coverage, assembling should not be a problem. Finally, blasting the EBV virus genome against assembled contigs should give a more correct integration site estimation because it considers the true host genome instead of a general human genome reference.

Availability of data: Are the deep sequencing reads (complete raw data or human host removed microbial part) available at any repository like EBI or NCBI?

Reviewer #3 (Remarks to the Author):

The authors Woerner et al. present their manuscript, "Circulating microbial content in myeloid malignancy patients carries diagnostic and prognostic potential," for review in which they examine deep coverage whole-genome DNA sequencing data from the blood and bone marrow of 1870 myeloid malignancy patients and 12 healthy donors for microbial DNA sequences. After careful filtering, they use the microbial reads to characterize the bacteria, fungi, and viruses they represent, finding associations between microbial signatures and myeloid malignancy diagnoses. A machine learning algorithm is used to classify myeloid disorder diagnoses based solely on microbial signatures.

Overall, the paper is well written and the methods clearly described. The topic is highly novel, building upon recent studies in solid tumors and AML datasets that show a similar disease specific microbial signatures detectable in the blood. This research raises additional questions about diagnostic, and perhaps, therapeutic applications as well as those that relate to the effects of therapy and supportive care on these signatures. And, it raises interest in the mechanisms associated with the different microbial signatures including whether the microbial shift might influence the development of disease instead of simply being a detectable epiphenomenon. Immediate clinical implications are unlikely as WGS is not routinely performed on patients nor are there commercial tests yet available for bloodborne microbial analysis. Furthermore, it is not clear that the prognostic or diagnostic associations found here are strong enough to improve upon current analytic methods. The conclusions from this necessarily descriptive study are appropriate but will need independent validation to ensure they are generalizable to independent populations and methods.

Comments and Suggestions

A glaring omission from the article is the lack of association between microbial signatures and mutation status. As the authors point out in the conclusion section, the pathogenic effects of certain mutations, like those in TET2, are associated with alteration in the gut microbiome in mice. It is certainly plausible that mutations associated with changes in gut permeability or bone marrow microenvironment in ways that enrich for particular microbes. Differences in mutation profiles between

myeloid disorders might partially explain the different microbial signatures observed here. Where there any such genetic associations with microbial signatures?

Even within diagnoses, mutation patterns can be unique and mutually exclusive. For example, JAK2 mutant MPN may be very different non-JAK2 mutant MPN. Was this explored?

The degree to which bloodborne microbial DNA can classify patients with myeloid neoplasms is somewhat surprising given the often ambiguous diagnostic boundaries between these disorders. The distinction between MDS and AML, for example, is an arbitrary threshold of 20% blasts and leukemic transformation does not represent a distinct biological process. Was there any association with microbial features that scaled with blast proportion? Was there a ceiling after which higher blasts counts no longer increased this signal?

Missing from the discussion is greater mention of the potential mechanisms for different microbial signatures in different disease states and what the potential confounders might be. For example, AML patients had the lowest alpha-diversity. Could this be because of greater neutropenia or greater likelihood of recent antibiotic use?

The number of controls is relatively small given the heterogeneity of the patient population they are being compared to. This includes differences in age, co-morbidities, prior exposures, diet, and medications taken which one could reasonably assume might impact the circulating microbiome as much or more than their myeloid malignancy. Perhaps a healthy control group is not the only control of interest here, since in practice, the clinical distinction of importance might be between malignant and benign causes of cytopenias. One would be curious if non-MDS causes of anemia, such as iron deficiency, thalassemia, hemolysis, or aplastic anemia might not be more difficult to discriminate from MDS, for example. These comparisons might give more insight into the mechanism of dysbiosis observed here.

The association between EBV reads and MDS disease risk is intriguing. Was there a component of the IPSS-R risk score that associated with EBV detection (e.g., anemia, chromosomal abnormality, blast proportion, thrombocytopenia)? Was there any association with eventual leukemic transformation (or subsequent response to therapy)?

It would be fascinating to know if patients with CHIP or CCUS have microbial signatures similar to those shown here and if these are more likely to resemble lower risk MDS, follow mutation specific patterns, or have unique signatures altogether. Questions raised by this study like this could be discussed.

Minor Points

The automated classifier appears to be determining if a patient has a particular diagnosis or not. It does not seem to be predicting which of the options is most likely? Was the test utilized in this way, and if so, how does it perform? Is 'Normal' an option for the automated classifier?

RESPONSE TO REVIEWER COMMENTS

We are grateful for each of these reviewers' insightful comments, which aided greatly in a much-improved revision.

Among other improvements, reviewer comments encouraged us to expand the analysis to examine relationships between microbial content and somatic mutations, as well as to consider associations between myeloblast percentage (which is used clinically to distinguish between MDS and AML) and microbial content. Intriguingly, we now find that myeloblast percentage likely drives the observed differences between MDS and AML microbiomes. We also now include qPCR validation of our *in silico* observations.

Below we reply to each referee comment, replicated here verbatim in **bold**, with our responses in *italics* immediately below each comment.

Reviewer #1 (Remarks to the Author):

1. Comparing 1,870 samples with 12 controls is highly problematic, particularly given the known diversity of the microbiome amongst healthy persons. All comparisons to healthy controls should be done with significant caution. Having a more robust group of healthy controls would strengthen such comparisons.

We would first like to point out that obtaining whole-genome sequence from healthy bone marrow samples is extremely difficult for obvious reasons, and we felt fortunate to be able to obtain even 12 samples that were processed and sequenced at the same facility as the cases were. This latter point is crucial, as it is well known that batch effects leading to artifactual conclusions are rife in microbiome studies, and therefore using public data as controls is quite challenging (please see further below in the response to this remark). Second, the main focus of our study is a description of the circulating microbial landscape of disease cases, and its relationship to clinical features (including specific diagnosis). The case-control comparisons are a smaller aspect of the manuscript, and indeed, as the reviewer recommends, we have been extremely cautious about drawing broad conclusions from the comparisons. The conclusions we do make, and the rationale that they are not invalidated by the small control sample size, are itemized further below in the response to this remark.

Nonetheless, we agree that the small number of controls is sub-optimal. To explore the feasibility of increasing the control sample size by using public controls, we acquired 50 whole-genome sequence files generated from the blood (public whole-genome sequence data from healthy bone marrow donors is not available, to our knowledge) of healthy subjects from NHLBI's Trans-Omics for Precision Medicine (TOPMed) study. We computed the principal coordinates from the genus relative abundances of all cases ($n = 1,870$), the 12 initial controls used in our study, and the TOPMed controls ($n = 50$). It can be seen in the figure immediately below that the TOPMed controls cluster far from the cases (all 1,870 cases are shown and overlap heavily with one another) and the 12 initial controls. On the other hand, our 12 initial controls are much more similar to the cases. The cases and 12 initial controls were all processed and sequenced at the same center.

This plot demonstrates the effect that different sequencing centers can have on studies of microbial content. It also demonstrates the relative uniformity in microbial content between our cases and 12 initial controls. Given the stark differences observed here between the TOPMed controls and our original cohort, it would be difficult if not impossible to distinguish the batch effects from true case/control differences. We were therefore left with only 12 normal controls. In recognition of this and of the Reviewer’s justifiable concern, we have now acknowledged the issue in the Discussion section, adding the following sentences:

“Finally, we acknowledge that our study has a relatively small number of healthy controls as compared to cases. Overall, we felt that the importance of using controls well-matched to cases in terms of tissue source and processing/sequencing site outweighed the statistical power gained by seeking other sources of data. Using externally-processed controls could lead to batch effects and artifactual results. As our results show, however, we were still well-powered to uncover a number of differences in microbial composition between the two groups”.

As stated above, however, the majority of our results do not rely on controls since the main focus is the microbial-clinical associations (especially diagnosis). We were extremely careful not to overinterpret the results derived from the small number of controls. We contend that virtually all (see below for the one exception) of the results initially reported remain valid despite this small number. Specifically, here we list all of the results involving the normal controls, and explain why our interpretations remain valid:

- Grouping of the controls in the tSNE plot (Figure 1b): *Even for a sample size of n=12, this grouping is almost certainly not random, as evidenced by P-value cited in the text ($P < 2.2 \times 10^{-16}$).*

- In terms of Bray-Curtis dissimilarity, controls are closer to one another than to cases (Figure 1c): To demonstrate that this observation is not tempered by the smaller sample size, we have now added a P-value ($P = 3.56 \times 10^{-40}$) to the Figure in the revision.
- Viral species are detected in only four of the 12 controls, and at low levels (Figure 2b): In the revision we have added a Wilcoxon test on the number of species-mapped viral reads in cases vs. controls, demonstrating ($P = 0.019$) that we are still powered to detect the differences.
- Lower levels of Proteobacteria in controls (Figures 3a and 3b): Again, despite having only 12 controls, we were adequately powered to detect this case/control difference in Proteobacteria levels ($P = 8 \times 10^{-7}$, as stated in the manuscript text).
- Smaller range of Proteobacterial levels and reduced presence of Firmicutes and Bacteroidetes in controls (last two sentences of first paragraph of “Landscape of the bacteriome in circulation” subsection in the original version): In this instance, the small number of controls may admittedly have had an impact. The smaller range and reduced presence of bacterial phyla in controls could indeed be the result of a much smaller number of controls as compared to cases. We have therefore removed these statements in the revision.
- Higher α -diversity in controls (Figure 3c): Once again, the low P-values (4.1×10^{-6} and 1.3×10^{-4}) indicate that the sample size left us adequately powered.

2. The samples were analyzed over a 12 year period which raises concern for introduction of various processing heterogeneity into the data. It is known that processing effects can have a major impact on microbiome studies so the authors should do an analysis looking at whether their microbial content changed in a temporal fashion.

We agree that temporal changes could be an issue. We therefore took the Reviewer’s suggestion and analyzed the data for relationships between time of diagnosis and microbial content among the cases. Indeed, we found clustering into four groups by time period: September 2005 - January 2011 ($n = 961$), February 2011 - November 2011 ($n = 234$), December 2011 - February 2013 ($n = 294$), and March 2013 - February 2017 ($n = 381$). This clustering is evident in the tSNE plot (see the first figure in the Supplementary Note in the resubmission).

We therefore next sought to determine whether our results are confounded by this temporal relationship. To this end, we have included with the resubmission an extensive Supplementary Note that repeats all of our original analyses, controlling for these temporal clusters, and demonstrating that virtually all of our conclusions remain valid, i.e., the conclusions of our original submission are *not* confounded by the temporal changes in microbial content. Indeed, it is likely that these temporal changes are in fact caused by the changes in relative frequencies of diagnoses within each time period. As we concluded in the original submission, microbial content is associated with diagnosis, and therefore it follows that temporal shifts in the relative frequencies of the four diagnoses would result in temporal shifts in microbial content. Indeed, our re-analysis in the Supplementary Note, this time controlling for time period, demonstrate that the results remain consistent with one exception: our observation that AML patients had the highest overall bacterial burden (very last statement in the first paragraph of the “Relationship between bacteriome and clinical characteristics” subsection in Results section of the original manuscript) did not hold up statistically when we controlled for temporal cluster. We have therefore removed this statement in the revised manuscript.

3. The samples were not processed initially to maximize microbial DNA but rather were focused on human DNA. Thus, the processing itself could have led to significant artifact in terms of what microbial DNA remained. This cannot be undone given the nature of the study but should be cited as a limitation of the study.

We agree with the reviewer's comment regarding this limitation of our study. We note, though, that we did cite this limitation in our original submission. Specifically, we had a paragraph in the Discussion section listing the weaknesses of our study, which contained the following sentences:

"Ideally, we would have included technical controls⁴⁶ in our study, to account for sources of contamination specific to our experimental protocols. Since the samples were originally collected in clinical practice for mutation profiling, such controls were not available. Nonetheless, our computational approach was conservative, omitting a large list of known reagent and kit contaminants along with other artifacts."

Here reference 46 cited an Eisenhofer, R., et al. (Trends in microbiology, 2019) study, entitled "Contamination in Low Microbial Biomass Microbiome Studies: Issues and Recommendations", that provides guidelines for optimally assay microbial DNA. To more clearly emphasize the reviewer's point that our data comes from samples processed for human DNA analysis, we have modified the language in the resubmission to state the following:

"Another issue is that our data was derived from samples processed for human DNA analysis. Ideally, we would have included technical controls⁵² in our study, to account for artifacts specific to our experimental protocols. Since the samples were originally collected in clinical practice for mutation profiling, such controls were not available. Nonetheless, our computational approach was conservative, omitting a large list of known reagent and kit contaminants along with other artifacts."

4. The authors should confirm some of their findings with targeted quantitative PCR. Given concerns for contamination and other mapping artifacts, targeted qPCR confirmation would be quite helpful. There are well established protocols for this for herpes viruses as well as various bacterial genera. Their finding that EBV presence can be used as a prognostic marker in low-intermediate risk MDS would particularly benefit from qPCR analysis.

Taking the Reviewer's suggestion, we performed qPCR validation on 18 MDS patient samples, nine in which EBV reads were found in the sequence data, and nine in which no EBV reads were found. The qPCR results were remarkably concordant with our original data. Among the nine patients without EBV reads, qPCR could not detect EBV in eight, and the ninth had the smallest estimated abundance among those with detectable EBV. qPCR was able to detect EBV in all nine that had EBV reads detected. Overall, the correlation between the sequence-based results and the qPCR results across all 18 samples was high (Spearman $\rho = 0.90$, $P = 3.4 \times 10^{-7}$). These findings are now reported in the main text and summarized in Supplementary Figure 4.

Specific comments:

1. Please check the y-axes on panel 1G. I do not see how the proportions add up to 1.

We apologize for the confusion and can understand how the bar graphs were somewhat ambiguous, leading to the confusion. The proportions were intended to represent the proportion of patients, within each cluster, that have the indicated cytogenetic lesion (NOT the proportion of patients with the cytogenetic lesion that fall within each cluster). To clarify, we have modified the horizontal axis label to state, "Karyotype frequency within each cluster", and have also changed the legend to state: "Barplots indicating proportion of patients, within each principal coordinate cluster, with complex karyotype, normal karyotype, and trisomy 8". We hope that these amendments clarify the graphical representation.

2. Although the authors claim their data suggests the ability to provide diagnostic criteria, in reality the AUCs are not strong. It is not realistic to think that a microbiome profile would be used to diagnose (and hence design treatments) for particularly myeloid malignancies. It is interesting that there are discriminative properties of their microbiome analyses but not to the point of using such data to diagnose.

We agree with the Reviewer that the AUCs are far from demonstrating that microbiome-based diagnosis is "clinic-ready". Our intent here was to show that the circulating microbiome does have a signal that tracks with diagnosis. We would also argue that, with further technological developments and biological insight, the potential of microbe-based diagnosis in the future is not terribly far-fetched. In the original submission, we were careful to qualify our claims with the word "potential".

Nonetheless, we see the Reviewer's point that we do not demonstrate the ability to precisely discriminate among diagnoses based on microbial content, and that care should be taken with claims made in the text. In the resubmission, therefore, we have tempered assertions of diagnostic potential throughout. For instance, we have even changed the title of the paper from "Circulating microbial content in myeloid malignancy patients carries diagnostic and prognostic potential" to "Circulating microbial content in myeloid malignancy patients has diagnosis-associated characteristics and prognostic potential".

3. For Figure 2, HHV-6 is well known to integrate into the germ-line of about 1% of humans leading to high-level viremia (see PMID 32044155). Probably this explains the high levels observed in some patients in Fig. 2A.

We appreciate the Reviewer's pointing this out, and we agree that this is the likely explanation. Interestingly, the HHV-6 burden that we observe in these patient outliers is roughly consistent with one copy of the HHV-6 genome per human cell, which is indeed consistent with germline integration. In the resubmission, we have now added the following passage:

"We observed extremely high levels of human betaherpesvirus 6 in nine (0.48%) patients. In these patients, the estimated number of copies of the viral genome approaches one per human cell. This observation is consistent with the known ability of human betaherpesvirus 6 to integrate into the human germline, which a recent study²³ found to occur in 0.58% of a cohort with primarily European ancestry. None of the other viruses showed evidence of germline integration."

4. It is not particularly surprising that bone marrow and blood samples from myeloid

malignancy patients would have high levels of EBV and CMV (to a lesser degree). These viruses are near universally present in humans and are latently present in hematologic cells with reactivation occurring during periods of immunosuppression. Although the authors touch on this in the discussion, these findings could more clearly be presented in the setting of what is known about these viruses.

We agree and have added the following sentences to the paragraph in our Results section that describes the presence of EBV/CMV:

“The higher abundance of viruses in some of the patients may be a result of disease-related immunosuppression. Indeed, both EBV and CMV are known to exist in a latent stage in large proportions of the population and can be reactivated upon suppression of host immune system²⁴.”

5. The y-axis of Fig. 2C read “Number of affected patients” whereas actually this is the number of patients samples in which viral reads are detected. There is no clinical data that these patients were “affected”.

We have changed the y-axis label in Figure 2c to “Patients with viral reads”.

6. Similarly, the authors write that “differences in viral infection...” on page 4. Infection implies some type of clinical issue. These viruses all establish latency in human cells so finding DNA does not imply infection. Similarly, the finding of reads mapping to *Trichosporon asahii* does not imply infection (page 5).

We agree and have changed the phrase “Differences in overall viral infection were observed...” to “Differences in overall viral read presence were observed...”, and the phrase “fungal infection was found...” to “fungal reads were found...”. We have also modified other instances of the word “infection” throughout the manuscript.

7. The rarefaction curve (Fig. 3E) is not discussed in the manuscript that I could find.

We do make reference to Figure 3e in the first paragraph of the “Relationship between bacteriome and clinical characteristics” subsection of the Results section in the original submission:

“AML patients also tended to have the lowest bacterial α -diversity (Figure 3c) and richness (Figure 3e) but, interestingly, the highest overall bacterial abundance (median 6048 vs. 3738 for non-AML cases; Wilcoxon $P < 2.2 \times 10^{-16}$)”.

8. On page 6, the authors state that *Dermabacter* and *Kytococcus* are opportunistic pathogens frequently affecting immunocompromised individuals. In reality, these organisms are exceedingly rare causes of infections.

We have modified the language in the resubmission to state:

*“...*Dermabacter* and *Kytococcus*, both of which have been reported to infect immunocompromised individuals”.*

9. In the discussion the authors write that they did not observe significant differences

between blood marrow and blood samples, but these data are not presented. It is important to do so because the samples are considered as a group (and typically the authors write about bone marrow samples when including both samples types).

In the original submission, the following sentence appears at the end of the first paragraph of the “Microbial landscape differs between cases and controls and among diagnoses” subsection of the Results section:

“Interestingly, in these t-SNE plots samples did not seem to group by age, sex, or blood/bone marrow status (Supplementary Figure 1), suggesting that these factors are not strongly associated with microbial content.”

Reviewer #2 (Remarks to the Author):

The work presents some interesting microbial candidates to discriminate tumor types and shows a diagnostic potential of the microbial composition. But individual microbial taxons could not be observed to work as a single strong marker that shows significant differences between disease types. Also the predictive power of the microbial total community has some diagnostics limits as healthy controls were not included in the classification analysis, and hence it’s a discrimination between disease types alone and not between healthy and disease.

The weakest point of this study is clearly the small number of 12 controls versus 1870 disease cases in the taxonomic analysis and a complete absence of healthy controls in the diagnostic analysis.

We would first like to point out that obtaining whole-genome sequence from healthy bone marrow samples is extremely difficult for obvious reasons, and we felt fortunate to be able to obtain even 12 samples that were processed and sequenced at the same facility as the cases were. This latter point is crucial, as it is well known that batch effects leading to artifactual conclusions are rife in microbiome studies, and therefore using public data as controls is quite challenging (please see further below in the response to this remark). Second, the main focus of our study is a description of the circulating microbial landscape of disease cases, and its relationship to clinical features (including specific diagnosis). The case-control comparisons are a smaller aspect of the manuscript, and indeed, as the reviewer recommends, we have been extremely cautious about drawing broad conclusions from the comparisons. The conclusions we do make, and the rationale that they are not invalidated by the small control sample size, are itemized further below in the response to this remark.

Nonetheless, we agree that the small number of controls is sub-optimal. To explore the feasibility of increasing the control sample size by using public controls, we acquired 50 whole-genome sequence files generated from the blood (public whole-genome sequence data from healthy bone marrow donors is not available, to our knowledge) of healthy subjects from NHLBI’s Trans-Omics for Precision Medicine (TOPMed) study. We computed the principal coordinates from the genus relative abundances of all cases ($n = 1,870$), the 12 initial controls used in our study, and the TOPMed controls ($n = 50$). It can be seen in the figure immediately below that the TOPMed

controls cluster far from the cases (all 1,870 cases are shown and overlap heavily with one another) and the 12 initial controls. On the other hand, our 12 initial controls are much more similar to the cases. The cases and 12 initial controls were all processed and sequenced at the same center.

This plot demonstrates the effect that different sequencing centers can have on studies of microbial content. It also demonstrates the relative uniformity in microbial content between our cases and 12 initial controls. Given the stark differences observed here between the TOPMed controls and our original cohort, it would be difficult if not impossible to distinguish the batch effects from true case/control differences. We were therefore left with only 12 normal controls. In recognition of this and of the Reviewer's justifiable concern, we have now acknowledged the issue in the Discussion section, adding the following sentences:

“Finally, we acknowledge that our study has a relatively small number of healthy controls as compared to cases. Overall, we felt that the importance of using controls well-matched to cases in terms of tissue source and processing/sequencing site outweighed the statistical power gained by seeking other sources of data. Using externally-processed controls could lead to batch effects and artifactual results. As our results show, however, we were still well-powered to uncover a number of differences in microbial composition between the two groups”.

As stated above, however, the majority of our results do not rely on controls since the main focus is the microbial-clinical associations (especially diagnosis). Furthermore, we were extremely careful not to overinterpret the results derived from the small number of controls. We contend that virtually all (see below for the one exception) of the results initially reported remain valid despite this small number. Specifically, here we list all of the results involving the normal controls, and explain why our interpretations remain valid:

- Grouping of the controls in the tSNE plot (Figure 1b): Even for a sample size of $n=12$, this grouping is almost certainly not random, as evidenced by P -value cited in the text ($P < 2.2 \times 10^{-16}$).
- In terms of Bray-Curtis dissimilarity, controls are closer to one another than to cases (Figure 1c): To demonstrate that this observation is not tempered by the smaller sample size, we have now added a P -value ($P = 3.56 \times 10^{-40}$) to the Figure in the revision.
- Viral species are detected in only four of the 12 controls, and at low levels (Figure 2b): In the revision we have added a Wilcoxon test on the number of species-mapped viral reads in cases vs. controls, demonstrating ($P = 0.019$) that we are still powered to detect the differences.
- Lower levels of Proteobacteria in controls (Figures 3a and 3b): Again, despite having only 12 controls, we were adequately powered to detect this case/control difference in Proteobacteria levels ($P = 8 \times 10^{-7}$, as stated in the manuscript text).
- Smaller range of Proteobacterial levels and reduced presence of Firmicutes and Bacteroidetes in controls (last two sentences of first paragraph of "Landscape of the bacteriome in circulation" subsection in the original version): In this instance, the small number of controls may admittedly have had an impact. The smaller range and reduced presence of bacterial phyla in controls could indeed be the result of a much smaller number of controls as compared to cases. We have therefore removed these statements in the revision.
- Higher α -diversity in controls (Figure 3c): Once again, the low P -values (4.1×10^{-6} and 1.3×10^{-4}) indicate that the sample size left us adequately powered.

Regarding the Reviewer's comment regarding diagnostic analysis, note that our approach relied on building a machine-learning classifier from training sets, then validating it on test sets. The training set comprised a randomly-selected 70% of the samples, with the remaining 30% comprising the test set (see Methods). Given only 12 healthy controls, this means that only 8 or 9 samples would be used to train the classifier, which would be evaluated on only 3 or 4. These numbers are far too small to either train a meaningful model or evaluate its performance. In contrast, our sample counts for the individual diagnoses AML, MDS, MDS/MPN, and MPN are 612, 640, 264, and 354, respectively, giving us ample sample sizes upon which to train the diagnostic classifiers.

Controls are also absent in PCoA plot Fig. 1e, but present in Fig. 1b. Why are healthy controls sometimes included and sometimes not?

As noted above, the primary focus of our study was the microbial landscape in myeloid malignancy patients and its relationship with clinical factors (indeed, our small number of controls precluded an extensive case-control study). As such, after briefly presenting data showing that the controls cluster together both in the tSNE plot (Fig 1b) and with regard to Bray-Curtis distance (Fig 1c), our presentation pivoted to its disease case focus. In particular, we consider relationships between microbial content and both diagnoses (Fig 1d-f) and karyotype (Fig 1g), both of which are irrelevant for healthy controls.

Another limiting factor is of course the amount of microbial sequences that could be extracted from bone and blood samples. Per sample counts of <4000 reads for bacteria and 120 for viruses are very small for reliable taxonomic identification.

To get a better picture of the microbial sequencing depth, it would be helpful to report not only the mean, but also the std, min, and max numbers of microbial related reads across all samples. And also what are the total numbers of raw read counts or base pairs per sample? Do the 12 control samples have a total number of raw reads comparable with disease samples (similar sequencing depth)?

To give a clearer idea of the raw and microbial read counts, we now provide a table (Supplementary Table 1) of total (human + microbial) read counts for each sample. As can be computed from this table, the total reads from normal samples (mean = 2,058,189,937, median = 2,014,504,780) is comparable to that from disease samples (mean = 2,131,806,786, median = 2,080,201,118). The differences between these two groups are significant using neither a *t*-test ($P = 0.2016$) nor a Wilcoxon test ($P = 0.4889$). The following figure shown the distribution of these read counts for the two groups:

Furthermore, as per the Reviewer's suggestion, we now report in the manuscript the mean, standard deviation, minimum and maximum numbers of microbial reads.

Taxonomic identification: While the use of PathSeq might work to count taxa related reads, the described need to exclude genomes having unbalanced or peaked converge regions could be avoided by using a taxonomic identifier like MetaPhlAn that does not simply count reads, instead claims a taxon to be present when hundreds of marker genes are present, hence forcing a more broad genome coverage. Or, alternatively, a kind of measure

like median coverage across all bases of a genome could be applied instead of error prone manual examination.

We agree that there are countless alternative paths we could have taken with regard to our data analysis and quality control. Our goal was to retain as much sensitivity to detect low-abundance taxa as possible, while minimizing false positives. This was considerably challenging given the nature of our low-biomass samples (blood and bone marrow). We did consider using MetaPhlAn (along with many other tools). Our main concern with MetaPhlAn was its requirement that high numbers of marker genes be present in order for a taxon to be deemed present. As the Reviewer notes, our read counts were often too low to give high depth coverage of individual species, and as such we concluded that MetaPhlAn would discard large numbers of taxa that are truly present. (This characteristic of MetaPhlAn has also been noted in the literature, for example Yap et al, "Evaluation of methods for the reduction of contaminating host reads when performing shotgun metagenomic sequencing of the milk microbiome", Scientific Reports 2020, among others). We also considered various coverage metrics of the sort the Reviewer suggests. However, we found the performance of such metrics to be highly variable, owing largely to the fact that species have hugely differing fractions of their genome that match portions of the human reference genome. As such, these matching sections would have no coverage, their reads having been filtered out as human in the upstream steps. This led to variable coverage of each species, and a one-size-fits-all metric inevitably yielded copious false negatives. In the end, therefore, we decided on manual inspection coupled with filtering of known contaminants. We realize that this approach is sub-optimal. However, we felt that, since this is the first large study of the circulating microbiome in myeloid malignancy patients, it was best to cast a wide net and clean the data post-hoc as accurately as possible, in the hope that subsequent studies using orthogonal methods will further validate our findings. Finally, we contend that the strong concordance between our read-based EBV presence/abundance estimates and the qPCR results newly presented in the revision (see new content in the "Human herpesviruses prevalent among myeloid malignancy patients, with prognostic implications in MDS" subsection of the Results section) argue favorably for our overall approach.

Figure 2a,b: Control samples may have a lower viral abundance, but comparing only the top highest 100 samples of 1870 disease samples with all controls is not correct. If 4 of 12 control samples (33%) carry viruses, Fig 2a has to show also 33% or >600 samples of case samples to have a fair visual comparison.

In the resubmission, we have revised the Figure 2a to show all viral burdens in 33% (624) of the 1870 disease cases as per the Reviewer's suggestion.

Figure 2c: Virus prevalence might be affected by the virus genome length. Working close to the detection limit (few virus related reads per sample), large virus genomes might be easier identified than small sized genomes. Is the genome size somehow considered or are there any corrections done?

The reviewer brings up a very good point here. In the original submission, we did not take genome length into consideration. In the resubmission, we now include adjustment for genome length. Briefly, we estimate what the observed prevalence of each viral species would be if it had the same genome size as the smallest detected virus (in our case, the Small anellovirus, also known

as the Torque teno midi virus, with genome length 2442). This can be estimated by statistically computing the probability, for each patient, that at least one of the reads from the virus would have aligned to a restricted 2442-base portion of its genome. Please see the Methods section for specific details of how these computations are performed. The prevalence estimates resulting from this modified analysis are now presented in Supplementary Figure 3.

Identifying virus integration sites: Yes, paired reads with one end mapping to the human genome and the other to virus genome, might be a hint of host integrated sequences. However, it's important to double check secondary read alignments in order to exclude random mapping on identical regions present in host and virus. This could be done by mapping these host-virus reads only to the human genome to confirm that only one read of a pair can be aligned to the host genome. The same mapping can be done using only the virus genome. Alternatively, instead of mapping against the hg19 human reference genome, the sample specific host genome could be reconstructed. Having a 100X human genome coverage, assembling should not be a problem. Finally, blasting the EBV virus genome against assembled contigs should give a more correct integration site estimation because it considers the true host genome instead of a general human genome reference.

Our search for EBV integration actually begins from the post-filtered reads, where the filter (as implemented by PathSeq) aggressively removes any reads that map to the host. We then query the post-filtered reads for singletons (i.e. those whose mate was filtered) that map to the EBV genome. It follows that the virus-mapped read of the pair has already been shown not to align to the host. On the other hand, the Reviewer is correct that this procedure does not confirm that the human-mapped read of the pair does not also map to the virus genome. Therefore, we now perform alignments of the human-mapped reads of each pair to the EBV genome. Our inspection of the result revealed three pairs whose evidence of human integration was questionable, and we have therefore removed these reads from the table of putative integration sites (Supplementary Table 3 in the resubmission) and have modified the language in the text describing the results.

Availability of data: Are the deep sequencing reads (complete raw data or human host removed microbial part) available at any repository like EBI or NCBI?

We have uploaded all human host-removed data to NCBI's Sequence Read Archive (SRA) repository (BioProject PRJNA746290 : Circulating Microbiome in Myeloid Malignancy). The data will be released upon publication, but a reviewer link is available, should the Reviewer like to access the raw data:

<https://dataview.ncbi.nlm.nih.gov/object/PRJNA746290?reviewer=1idt3sd3e51hbhgh62d1demake>

Reviewer #3 (Remarks to the Author):

A glaring omission from the article is the lack of association between microbial signatures and mutation status. As the authors point out in the conclusion section, the pathogenic effects of certain mutations, like those in TET2, are associated with alteration in the gut

microbiome in mice. It is certainly plausible that mutations associated with changes in gut permeability or bone marrow microenvironment in ways that enrich for particular microbes. Differences in mutation profiles between myeloid disorders might partially explain the different microbial signatures observed here. Where there any such genetic associations with microbial signatures?

The Reviewer brings up a very good point here, and in the revision we now investigate the relationships between mutations and microbial content. To this end, we queried the literature for the most frequently mutated gene in AML, MDS, MDS/MPN, and MPN, and in the end focused on 24 different mutations: (23 genes, with FLT3-ITD treated distinctly from other FLT3 mutations): ASXL1, CALR, CBL, CEBPA, DNMT3A, EZH2, FLT3, FLT3-ITD, IDH1, IDH2, JAK2, KRAS, NPM1, NRAS, RUNX1, SETBP1, SF3B1, SRSF2, STAG2, TET2, TP53, U2AF1, WT1, and ZRSR2. Each of these genes is mutated in at least 2% of the cohort. We then tested whether the mutation status in any of these genes is associated with bacterial characteristics, specifically overall bacterial burden, genus α -diversity, Proteobacteria relative abundance, and Firmicutes relative abundance + Bacteroidetes relative abundance (this last characteristic was used as a proxy for potential gut permeability, since these Firmicutes and Bacteroidetes are the two dominant phyla in gut microflora). Three of the association tests achieved Bonferroni-corrected statistical significance: DNMT3A mutations were associated with lower genus α -diversity (logistic regression $P = 0.00027$; Figure 6a), while FLT3 and NPM1 mutations were associated with higher Proteobacteria levels (logistic regression $P = 0.00052$ and 0.00030 , respectively; Figure 6b,c). These results are now included in the new “Relationship between bacteriome and host mutations” subsection of the Results section.

Even within diagnoses, mutation patterns can be unique and mutually exclusive. For example, JAK2 mutant MPN may be very different non-JAK2 mutant MPN. Was this explored?

In the resubmission, we have now compared bacterial characteristics between JAK2-mutant and JAK2-wild type MPN patients. We found no statistical difference between the two groups in any of the four microbial measures mentioned just above. However, we found that a microbe-based machine-learning classifier could distinguish between the two groups better than chance, indicating some relationship between JAK2 mutation status and circulating microbiome in MPN patients. These results are now included in the manuscript in the new “Relationship between bacteriome and host mutations” subsection of the Results section.

The degree to which bloodborne microbial DNA can classify patients with myeloid neoplasms is somewhat surprising given the often ambiguous diagnostic boundaries between these disorders. The distinction between MDS and AML, for example, is an arbitrary threshold of 20% blasts and leukemic transformation does not represent a distinct biological process. Was there any association with microbial features that scaled with blast proportion? Was there a ceiling after which higher blasts counts no longer increased this signal?

Taking the Reviewer’s suggestion, we have now examined the relationship between blast counts and microbe content in two general ways. First, in the original manuscript, there were specific microbial characteristics that differed between AML and MDS. We tested whether each of these

characteristics scaled with blast proportion (in the expected direction), and indeed all of them did. Specifically: 1) Individuals with viral reads detected had statistically lower blast percentage, mirroring our original observation that viral reads were detected in a higher proportion of MDS vs. AML patients; 2) Blast percentage was significantly negatively correlated with viral burden, mirroring our original observation that MDS patients had the highest viral burden; 3) Blast percentage was significantly positively correlated with Proteobacteria relative abundance, mirroring our observation that AML patients had the highest Proteobacteria relative abundance; and 4) Blast percentage was significantly negatively correlated with α -diversity on most taxonomic levels, mirroring our original observation of lower α -diversity for AML patients. In the resubmission, we now include these results in a new subsection "Microbial differences between AML and MDS is largely driven by myeloblast percentage" of the Results section.

Second, also included in the new subsection is a search for the blast count threshold at which the machine learning classifier can optimally distinguish between patients above and below the threshold. Intriguingly, the optimum (i.e. the threshold for which the classifier attains the highest AUROC) is 18%, quite close to the clinical threshold of 20%. Please see the main text for details on these analyses.

Missing from the discussion is greater mention of the potential mechanisms for different microbial signatures in different disease states and what the potential confounders might be. For example, AML patients had the lowest alpha-diversity. Could this be because of greater neutropenia or greater likelihood of recent antibiotic use?

We agree that the factors mentioned by the Reviewer could certainly be confounders. Although we do not have data on antibiotic use, we do have absolute neutrophil counts (ANC) for 1002 of the patients. We find correlation between ANC and some microbial characteristics (e.g. some α -diversity measures) in the overall cohort, but this is likely due to both metrics (ANC and α -diversity) varying by diagnosis.

As the Reviewer suggests, we now add more discussion on potential mechanisms and confounders, specifically:

"Hematopoietic cell composition in circulation could also affect microbial content. AML patients have a higher prevalence of neutropenia, and absolute neutrophil counts (ANC) vary greatly by diagnosis. In our data, although ANC does correlate with some microbial features (e.g. α -diversity), we found no evidence that this correlation holds within diagnostic categories (data not shown). Alternatively, specific microbes or combinations of microbes could conceivably contribute to leukemic transformation or other clinical aspects. Given the relationships between MDS and EBV reported here, we tested for association between EBV and eventual leukemic transformation but found none (data not shown). Another possibility is that the observed associations could be the result of confounding factors. For instance, differences in antibiotic use among the different diagnoses would produce diagnosis-specific microbial signatures. We do not have data on patient medications, but it is important to note that all samples analyzed here were taken at diagnosis, and any antibiotics differentially prescribed because of diagnosis would not yet have affected these signatures."

As another potential confounding issue, we also point out in the revision that our newly-reported associations between FLT3/NPM1 mutations and Proteobacteria abundance may be confounded by the fact that both track with diagnosis.

The number of controls is relatively small given the heterogeneity of the patient population they are being compared to. This includes differences in age, co-morbidities, prior exposures, diet, and medications taken which one could reasonably assume might impact the circulating microbiome as much or more than their myeloid malignancy. Perhaps a healthy control group is not the only control of interest here, since in practice, the clinical distinction of importance might be between malignant and benign causes of cytopenias. One would be curious if non-MDS causes of anemia, such as iron deficiency, thalassemia, hemolysis, or aplastic anemia might not be more difficult to discriminate from MDS, for example. These comparisons might give more insight into the mechanism of dysbiosis observed here.

We agree that the number of controls here is sub-optimal. We were restricted to these controls because, in an effort to avoid artifacts that would likely emerge by using controls processed at a different sequencing facility, we exclusively used data from samples processed/sequenced at the same facility as our controls. Naturally, whole-genome sequence from healthy bone marrow donors is a rare commodity, and therefore these restrictions left us with the small number of controls (though we felt fortunate that we were able to obtain these). On the other hand, our study primarily centers on the description of the circulating microbial landscape of disease cases, and its relationship to clinical features (including specific diagnosis). The case-control comparisons are a smaller aspect of the manuscript, and we have been extremely cautious about drawing broad conclusions from the comparisons. For a more detailed discussion of these points, please see our responses above to Reviewer comments regarding the small number of controls.

With regard to the Reviewer's suggestions for alternative controls, we agree that such comparisons would be excellent bases for follow-up studies. For the current study, however, we lack whole-genome sequence from individuals with non-MDS anemia. We hope that such data will become increasingly available in the future.

The association between EBV reads and MDS disease risk is intriguing. Was there a component of the IPSS-R risk score that associated with EBV detection (e.g., anemia, chromosomal abnormality, blast proportion, thrombocytopenia)? Was there any association with eventual leukemic transformation (or subsequent response to therapy)?

We tested for association between the IPSS-R risk score components and EBV detection in MDS patients. There was only one nominal association observed (thrombocyte levels are slightly higher in patients without EBV detected, $P = 0.033$), a P -value that would not withstand multiple-test correction. This result reinforces that EBV detection is likely a marker of risk that is independent of IPSS-R, consistent with our observation that EBV presence/absence refines risk assessment in IPSS-R low individuals. In the resubmission, these analyses are reported as follows:

"To determine more directly whether the impact of EBV on survival is related to the clinical components of the IPSS-R score (hemoglobin, neutrophils, platelets, blasts, and cytogenetics), we tested for association between these components and EBV detection in MDS patients. None of the associations are significant, save that platelet levels are nominally significantly lower in EBV-positive individuals (Wilcoxon $P = 0.033$). This overall lack of association suggests that EBV

presence is a risk factor that is independent of IPSS-R, which is unsurprising given our observation that EBV refines IPSS-R's low-risk prognosis."

We observed no association between EBV and either leukemic transformation or subsequent response to therapy.

It would be fascinating to know if patients with CHIP or CCUS have microbial signatures similar to those shown here and if these are more likely to resemble lower risk MDS, follow mutation specific patterns, or have unique signatures altogether. Questions raised by this study like this could be discussed.

We agree, and have added the following verbiage to the Discussions section:

"A natural extension of the work presented here would be to query the circulating microbiomes of patients with MDS precursor conditions such as clonal hematopoiesis of indeterminate potential (CHIP) and clonal cytopenia of undetermined significance (CCUS). Given the differences in microbial characteristics that we observed among different diagnoses, risk categories, and gene mutation statuses, one might hypothesize that individuals with CHIP or CCUS would show microbial signatures that are intermediary between normal and MDS (particularly low-risk MDS), and that the CHIP/CCUS-defining mutation(s) may track with microbial characteristics. Such studies would shed further light on the implications and potential mechanisms of the associations reported in the current work."

Minor Points

The automated classifier appears to be determining if a patient has a particular diagnosis or not. It does not seem to be predicting which of the options is most likely? Was the test utilized in this way, and if so, how does it perform? Is 'Normal' an option for the automated classifier?

We initially constructed only binary classifiers (one diagnosis vs. the rest) because it is easier to assess these using the ROC curve (sensitivity vs. FDR) with two categories predicted. However, we have subsequently learned that the community commonly uses a method from "A Simple Generalisation of the Area Under the ROC Curve for Multiple Class Classification Problems" by Hand and Till (Machine Learning, 2001) to assign an AUROC measure to multiple-class classifiers. In the resubmission, we now also include a machine-learning classifier that aims to assign one of the four diagnoses to each patient based on microbial content, and we report the AUROC values based on the method of Hand and Till.

Regarding the Reviewer's comment question regarding the inclusion of "Normal" in the automated classifier, note that our approach relied on building a machine-learning classifier from training sets, then validating it on test sets. The training set comprised a randomly-selected 70% of the samples, with the remaining 30% comprising the test set (see Methods). Given only 12 healthy controls, this means that only 8 or 9 samples would be used to train the classifier, which would be evaluated on only 3 or 4. These numbers are far too small to either train a meaningful model or evaluate its performance. In contrast, our sample counts for the individual diagnoses AML,

MDS, MDS/MPN, and MPN are 612, 640, 264, and 354, respectively, giving us ample sample sizes upon which to train the diagnostic classifiers.

REVIEWER COMMENTS

Reviewer #1 (Remarks to the Author):

The authors have carefully responded to the reviewer's concerns and suggestions. I have no further concerns or suggestions.

Reviewer #2 (Remarks to the Author):

The authors have addressed my main concerns sufficiently and made the necessary changes in their sequencing analysis.

However, it remains a misleading text related to the low number of 12 healthy samples. The introduction reads like the main aim of their study would be to compare between healthy and disease, for example: "shotgun sequencing of DNA in the bone marrow and blood of 1,870 myeloid malignancy patients, as well as in the bone marrow of healthy donors". It would help the reader's understanding a lot, if instead hidden in the discussion, already the introduction or the beginning of the result section would contain some sentences that the main focus is on discriminating between disease subtypes and that the comparison with healthy individuals has some limits for drawing conclusions.

Reviewer #3 (Remarks to the Author):

The authors Woerner et al have revised their manuscript now entitled, "Circulating microbial content in myeloid malignancy patients has diagnosis-associated characteristics and prognostic potential," for resubmission. They have addressed the bulk of reviewer comments and included novel analyses that enhance confidence in their conclusions. In addition, they better address limitations of the study and avoid overstatement. These results are in line with recent findings in cancer patients that circulating microbiome patterns detected in blood samples prepared for DNA sequencing highly correlate with tumor type across a broad spectrum of malignancies. Associations with specific mutations and blast composition are of interest and may suggest potential mechanisms associated with microbiome variation. I have no further comments or suggestions.

RESPONSE TO REVIEWER COMMENTS

Below we reply to each referee comment, replicated here verbatim in **bold**, with our responses in *italics* immediately below each comment.

Reviewer #1 (Remarks to the Author):

The authors have carefully responded to the reviewer's concerns and suggestions. I have no further concerns or suggestions.

We are pleased that we have adequately addressed the Reviewer's concerns/suggestions.

Reviewer #2 (Remarks to the Author):

The authors have addressed my main concerns sufficiently and made the necessary changes in their sequencing analysis.

We are pleased that we have sufficiently addressed the Reviewer's main concerns.

However, it remains a misleading text related to the low number of 12 healthy samples. The introduction reads like the main aim of their study would be to compare between healthy and disease, for example: "shotgun sequencing of DNA in the bone marrow and blood of 1,870 myeloid malignancy patients, as well as in the bone marrow of healthy donors". It would help the reader's understanding a lot, if instead hidden in the discussion, already the introduction or the beginning of the result section would contain some sentences that the main focus is on discriminating between disease subtypes and that the comparison with healthy individuals has some limits for drawing conclusions.

In the new revision, we have now added, at the end of the Introduction, the following sentences clarifying that the main focus is on discriminating between disease subtypes, and that the small number of healthy controls limits conclusions to be drawn from their comparison with disease cases:

"Our primary goal was to elucidate relationships between microbial content/abundance and clinical features, including diagnosis and outcomes, in the disease cohort. Although our ability to perform case/control comparisons was limited by a relatively small number of healthy controls, we were nonetheless able to observe some significant differences in microbial content between patients and healthy donors."

Reviewer #3 (Remarks to the Author):

The authors Woerner et al have revised their manuscript now entitled, "Circulating microbial content in myeloid malignancy patients has diagnosis-associated characteristics and prognostic potential," for resubmission. They have addressed the bulk of reviewer comments and included novel analyses that enhance confidence in their conclusions. In addition, they better address limitations of the study and avoid overstatement. These results are in line with recent findings in cancer patients that

circulating microbiome patterns detected in blood samples prepared for DNA sequencing highly correlate with tumor type across a broad spectrum of malignancies. Associations with specific mutations and blast composition are of interest and may suggest potential mechanisms associated with microbiome variation. I have no further comments or suggestions.

We are pleased to have addressed all of the Reviewer's comments/suggestions.